# Optimal features for auditory categorization

Shi Tong Liu[1], Pilar Montes-Lourido[2], Xiaoqin Wang [3] & Srivatsun Sadagopan [1,2,4]

Humans and vocal animals use vocalizations to communicate with members of their species. A necessary function of auditory perception is to generalize across the high variability inherent in vocalization production and classify them into behaviorally distinct categories ('words' or 'call types'). Here, we demonstrate that detecting mid-level features in calls achieves production-invariant classification. Starting from randomly chosen marmoset call features, we use a greedy search algorithm to determine the most informative and least redundant features necessary for call classification. High classification performance is achieved using only 10–20 features per call type. Predictions of tuning properties of putative feature-selective neurons accurately match some observed auditory cortical responses. This feature-based approach also succeeds for call categorization in other species, and for other complex classification tasks such as caller identification. Our results suggest that high-level neural representations of sounds are based on task-dependent features optimized for specific computational goals.

[1] Department of Bioengineering, University of Pittsburgh, Pittsburgh 15213 PA, USA. [2] Department of Neurobiology, University of Pittsburgh, Pittsburgh 15213 PA, USA. [3] Department of Biomedical Engineering, Johns Hopkins University, Baltimore 21205 MD, USA. [4] Department of Otolaryngology, University of Pittsburgh, Pittsburgh 15213 PA, USA. Correspondence and requests for materials should be addressed to S.S. (email: vatsun@pitt.edu)

Human speech recognition is a highly robust behavior, showing tolerance to variations in prosody, stress, accents, and pitch. For example, speech features such as formant frequencies exhibit large variations within- and between-speakers[1,2], arising from production mechanisms (production variability). To achieve accurate speech recognition, the auditory system must generalize across these variations. This challenge is not uniquely human. Animals produce species-specific vocalizations (calls) with large within- and between-caller variability[3], and must classify these calls into distinct categories to produce appropriate behaviors. For example, in common marmosets (*Callithrix jacchus*), a highly vocal New World primate species, critical behaviors such as finding other marmosets when isolated depend on accurate extraction of call-type and caller information[4–8]. Similar to human speech, marmoset call categories overlap in their long-term spectra (Fig. 1a), precluding the possibility that calls can be classified based on spectral content alone, and requiring selectivity for fine spectrotemporal features to classify calls. At the same time, marmoset calls also show considerable production variability along a variety of acoustic parameters[8]. For example, twitter calls produced by different marmosets vary in such parameters as dominant frequencies, lengths, inter-phrase intervals, and harmonic ratios (Fig. 1). Tolerance to large variations in spectrotemporal features within each call type is thus necessary to generalize across this variability. Therefore, there is a simultaneous requirement for fine and broad selectivity for production-invariant call classification. The present study explores how the auditory system resolves these conflicting requirements.

This problem of requiring fine- and tolerant feature tuning, necessitated by high variability amongst members belonging to a category, is not unique to the auditory domain. For example, in visual perception, object categories such as faces also possess a high degree of intrinsic variability[9–12]. To classify faces from other objects, using an exemplar face as a template typically fails because this does not generalize across within-class variability[12]. Face detection algorithms use combinations of mid-level features, such as regions with specific contrast relationships[13,14], or combinations of face parts[12], to accomplish classification. Of these algorithms, the one proposed by Ullman et al.[12] is especially interesting because of its potential to generalize to other classification tasks across sensory modalities. In this algorithm, starting from a set of random fragments of faces, the authors used greedy search to extract the most informative fragments that were highly conserved across all faces despite within-class variability. Post hoc analyses revealed that these fragments were mid-level, i.e., they typically contained combinations of face parts, such as eyes and a nose. The features identified using this algorithm were consistent with some physiological observations, for example at the level of BOLD responses[15]. While the differences between visual and auditory processing are vast, these results inspired us to ask whether a similar concept – sound categorization using combinations of acoustic features – could be implemented by the auditory system.

The behavioral salience of calls for marmosets[4–8], and the increasing resources allocated to the processing of calls along the cortical processing hierarchy[16], suggest that call processing is a computational goal of auditory cortex. Call processing involves detecting the presence of calls in the acoustic input, classifying them into behaviorally relevant categories, extracting information about caller identity, determining the behavioral state of the caller, and developing situational awareness of the environment. Although a number of studies have described call-selective responses at various stages of the auditory pathway, there has

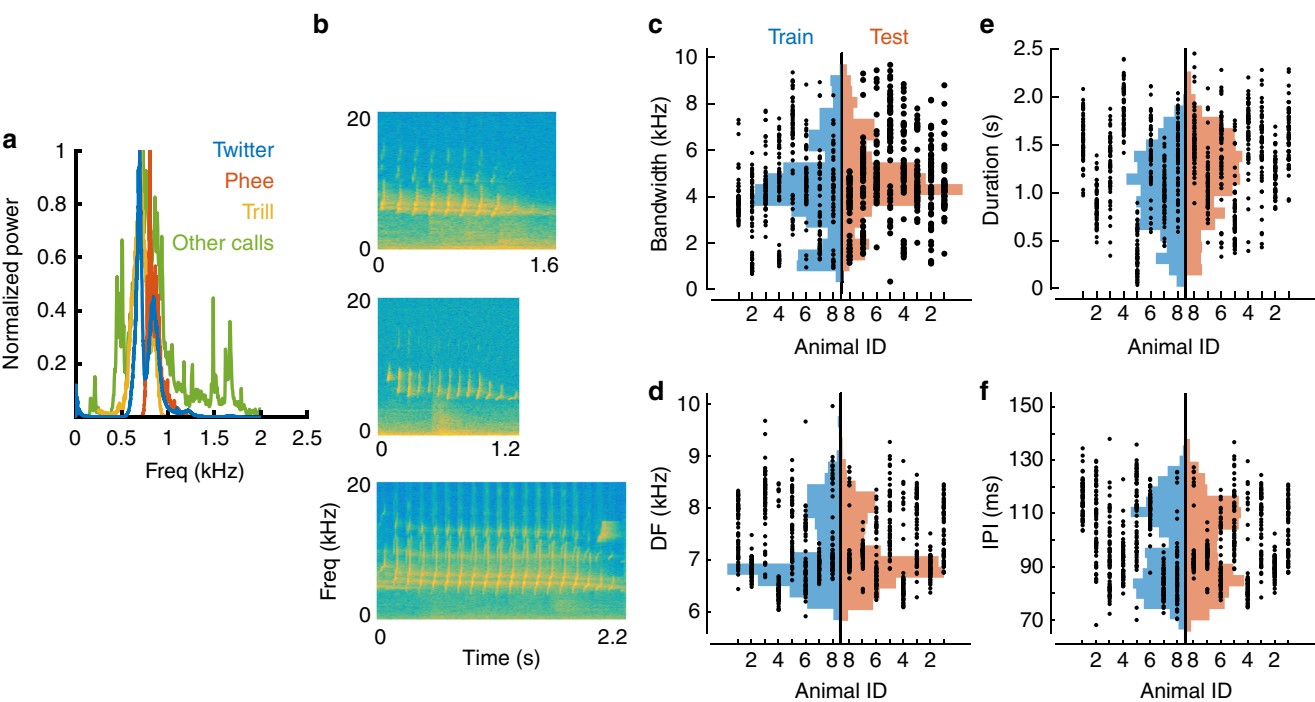

**Fig. 1** Production variability in marmoset calls. **a** The overall spectra of 3 major marmoset call types and other minor call types (grouped as 'Other calls'), showing spectral overlap between call categories. **b** Spectrograms of three twitter calls showing examples of production variability between individuals. **c–f** Production variability of twitter calls quantified along multiple parameters: **c** bandwidth, **d** dominant frequency, **e** duration, and **f** inter-phrase interval. Dots are parameter values of a single call produced by an individual marmoset. Histograms are overall parameter distributions, split into the training (blue) and testing (red) sets. These data show the large production variability captured by the training and test datasets, over which the model must generalize. No systematic bias is evident in calls used for model training and testing

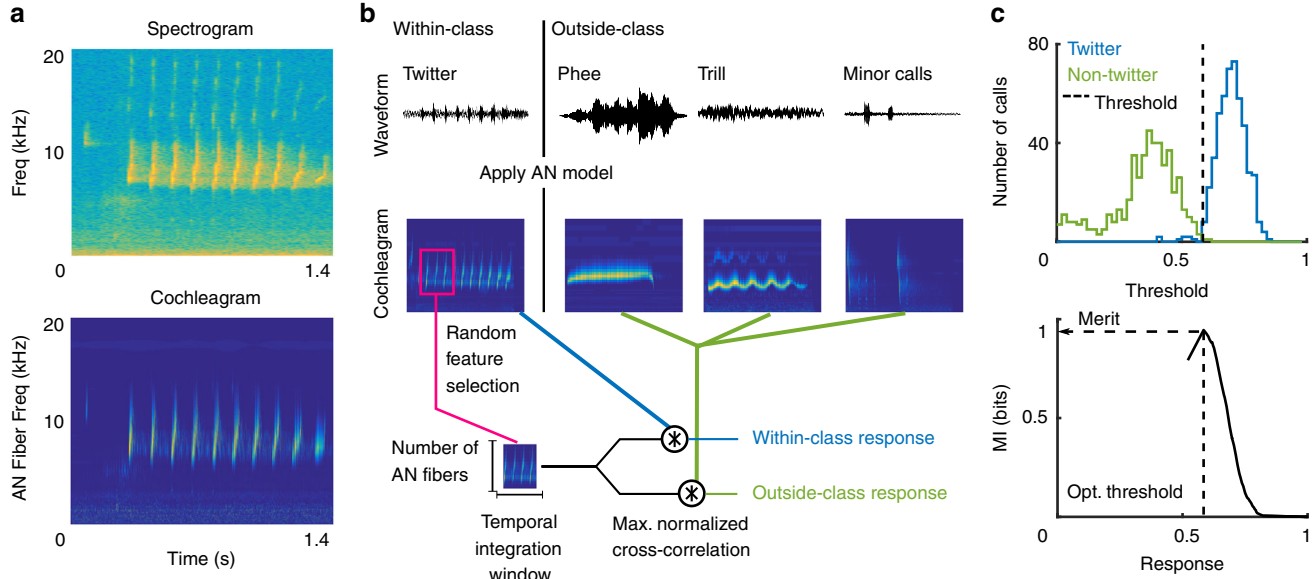

**Fig. 2** Initial feature generation and evaluation. **a** The spectrogram of a twitter call (top), and its corresponding cochleagram (bottom). Cochleagram color scale denotes firing rates of auditory nerve fibers. **b** Schematic for initial random feature generation for a twitter (within-class) versus other calls (outside-class) categorization task. Waveforms (top) were converted to cochleagrams (middle). The magenta box outlines a random initial feature picked from the twitter cochleagram shown. The maximum value of the normalized cross-correlation function between each call (within-class—blue, outside-class—green) and each random feature was taken to be the response of a feature to a call. **c** Distributions (top) of a feature's responses to 500 within-class (blue) and 500 outside-class (green) calls. The mutual information (bottom) of a feature computed as a function of a parametrically varied threshold. The dotted line, corresponding to maximal mutual information, is taken to be each feature's optimal threshold

been little investigation into how the auditory system goes about solving these problems, both at the algorithmic and mechanistic levels. In this study, we start with the premise that the detection and classification of calls into discrete call types is a critical first step that enables the above computations. Our overall question in this study is to ask how production-invariant call classification can be accomplished in the auditory pathway. Specifically, we test the hypothesis that production-invariant call classification can be accomplished by detecting constituent features that maximally distinguish between call types. Starting from an initial set of randomly selected marmoset call features, we use a greedy search algorithm to determine the most informative and least redundant set of features necessary for call classification. We show that high classification performance can indeed be achieved by detecting combinations of a small number of mid-level features. We then demonstrate that predictions of tuning properties of putative feature-selective neurons match previous data from marmoset primary auditory cortex. Finally, we show that the same algorithm is equally successful in caller identification with marmoset calls, and in call classification in other species such as guinea pigs (*Cavia porcellus*) and macaque monkeys (*Macaca mulatta*). Taken together, our findings suggest that classification of sound categories using mid-level features may be a general auditory computation.

## Results

**Intermediate features are more informative for classification**. We start with the premise that the first step in call processing is the categorization of calls into discrete call types, generalizing across the production variability that is inherent to calls. Let us consider the example of classifying twitter calls from all other call types. Marmoset twitters can be characterized along several acoustic parameters, such as bandwidth, duration, dominant frequency, and inter-phrase interval[8]. In Fig. 1c–f, we plot the

values of these parameters for individual calls emitted by 8 animals, showing the extent of within- and between-individual variability over which generalization is required for twitter categorization. Similar generalization is required for categorizing the other call types as well (Supplementary Fig. 1). We first generated 6000 random initial features from the cochleagrams of 500 twitter calls emitted by 8 marmosets ('training' set, blue histograms in Fig. 1). For the purposes of this study, a feature is a randomly selected rectangular segment of the cochleagram, corresponding to the spatiotemporal activity pattern of a subset of auditory nerve fibers within a specified time window. For each random feature, we determined an optimal threshold at which its utility for classifying twitters from other calls was maximized. The merit of each feature was taken to be the mutual information value (in bits) at this optimal threshold (Fig. 2; Equation 1).

In Supplementary Fig. 2, we plot the merits of all 6000 initial features as a function of each feature's bandwidth and temporal integration window. Along the margins, we plot the maximum merit of features within each bandwidth- or temporal window bin. These distributions compare the best features from each time bin, and show that features of intermediate lengths relative to the total call length show higher merits for call classification. This is an expected consequence of two characteristics of calls: (1) call types overlap in spectral content, so that brief features do not contain sufficient information to separate out categories, and (2) calls have high production variability, so that long features are less likely to be found across all calls belonging to the same category. We observed similar distributions for the classification of other marmoset call types, i.e., for trill vs. other calls, and phee vs. other calls (Supplementary Fig. 2). We then characterized feature complexity using a kurtosis-based metric (Methods). While features of low merit showed low complexity values and whole calls showed high complexity values, features of high-merit showed intermediate complexity values. This observation supported the hypothesis that mid-level features of

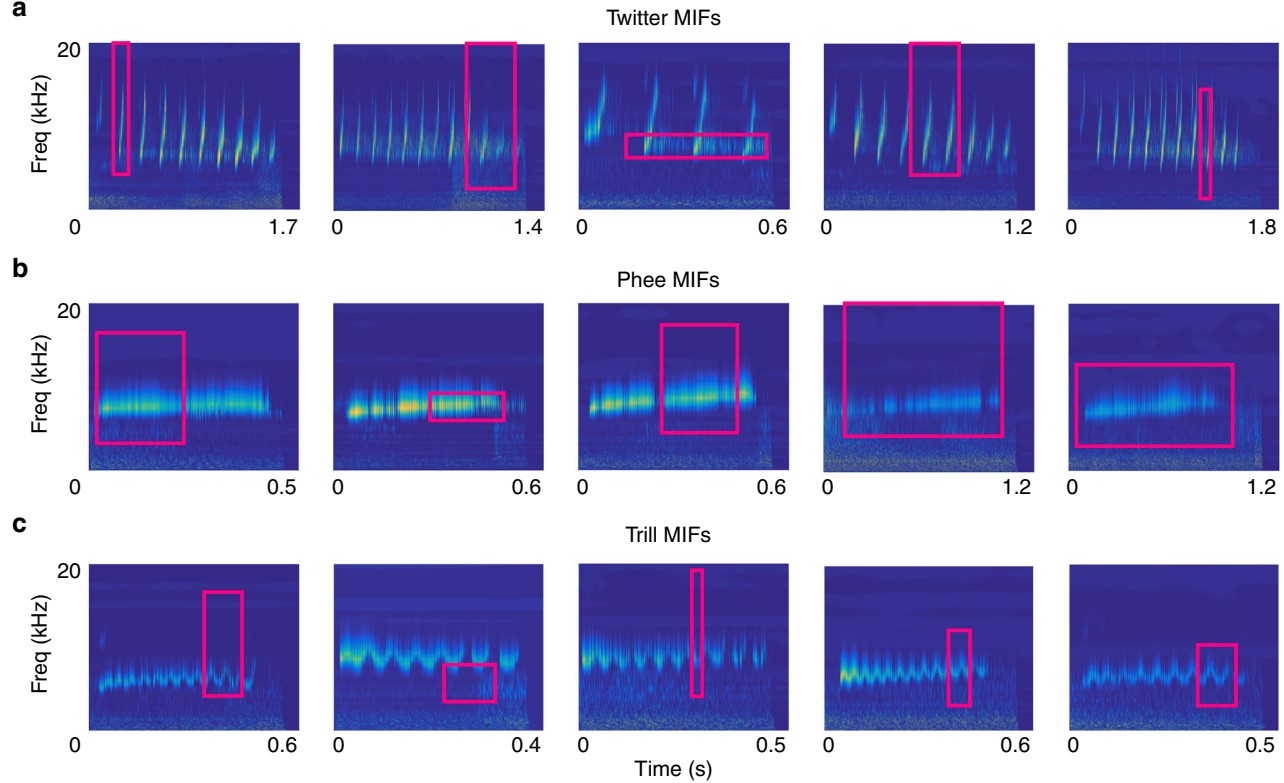

**Fig. 3** Most informative features for the classification of marmoset calls. Magenta boxes correspond to MIFs for the classification of **a** twitters vs. all other calls, **b** phees vs. all other calls, and **c** trills vs. all other calls, overlaid on the cochleagrams of the parent calls from which the MIFs were obtained

**Table 1 Information content of MIFs**

| | Twitter | | | Phee | | | Trill | | |
|---|---|---|---|---|---|---|---|---|---|
| MIF # | Added Info. | Merit | Weight | Added Info. | Merit | Weight | Added Info. | Merit | Weight |
| 1 | 0.95 | 0.95 | 14.58 | 0.78 | 0.78 | 10.06 | 0.60 | 0.60 | 7.88 |
| 2 | 0.01 | 0.84 | 12.14 | 0.01 | 0.67 | 7.76 | 0.10 | 0.12 | 5.37 |
| 3 | 0.01 | 0.44 | 9.26 | 0.01 | 0.74 | 8.65 | 0.04 | 0.12 | 4.40 |
| 4 | 0.01 | 0.85 | 12.49 | 0.01 | 0.71 | 8.29 | 0.04 | 0.25 | 7.13 |
| 5 | 0.01 | 0.87 | 12.49 | 0.01 | 0.75 | 8.87 | 0.04 | 0.53 | 7.59 |
| 6 | <0.01 | 0.87 | 12.49 | 0.01 | 0.72 | 8.39 | 0.03 | 0.43 | 6.18 |
| 7 | <0.01 | 0.80 | 11.71 | 0.01 | 0.71 | 8.27 | 0.03 | 0.29 | 7.44 |
| 8 | <0.01 | 0.84 | 12.30 | <0.01 | 0.71 | 8.27 | 0.03 | 0.27 | 8.14 |
| 9 | <0.01 | 0.39 | 8.97 | <0.01 | 0.75 | 8.90 | 0.02 | 0.27 | 8.26 |
| 10 | <0.01 | 0.34 | 8.62 | <0.01 | 0.71 | 8.49 | 0.02 | 0.22 | 7.74 |

The added information, merit, and weight (log-likelihood ratio) of the top 10 MIFs for twitter, phee, and trill

intermediate complexity were most informative for classification (Supplementary Fig. 2).

**Most informative features for classification.** Because, we generated the initial features at random, many of these have low merit, and many are similar. Therefore, the set of optimal features for classification is expected to be much smaller than this initial set. To determine the set of optimal features that together maximize classification performance, we used a greedy-search algorithm (see Methods). Briefly, we started with the feature of highest merit, and successively added features that maximized pairwise mutual information with respect to the already chosen feature set. We refer to the set of these optimal features as most informative features (MIFs) following the nomenclature of Ullman et al.[12,17]. We determined that call classification could be

accomplished using 11 MIFs for twitter vs. all other calls, 20 MIFs for trill vs. all other calls, and 16 for phee vs. all other calls. In Fig. 3, magenta boxes outline the top 5 MIFs that are optimal for each of these classification tasks (the first five MIFs in Table 1). The optimal features that we arrive at are mostly intuitive – for example, the top MIFs for classifying twitters detect the frequency contour of individual twitter phrases and the repetitive nature of the twitter call. In some cases, features seemed counter-intuitive— for example, the second MIF for trill classification seems to detect empty regions of the cochleagram. In this theoretical framework, the lack of energy at those frequencies is also informative about the presence of a trill.

In Table 1, we show the pairwise information added by each MIF, the merits, and the weights of the top 10 MIFs for these classification tasks. Note that 1 bit of information corresponds to

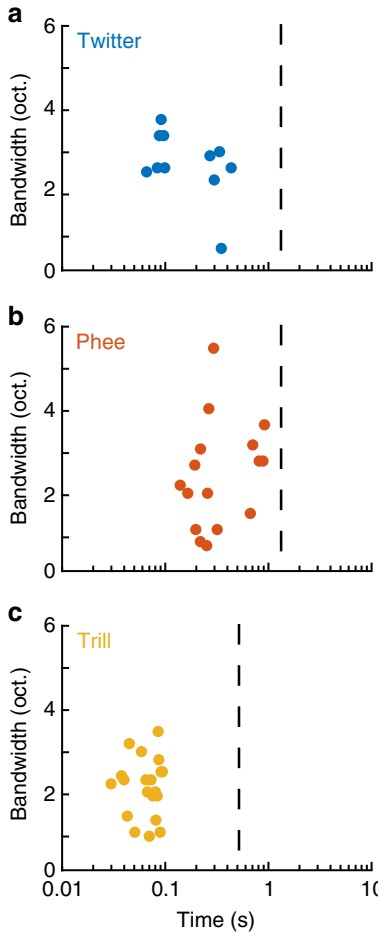

**Fig. 4** MIFs are of intermediate bandwidths and lengths. Scatter plot of the distribution of all MIFs for **a** twitters, **b** phees, and **c** trills as a function of their bandwidth and temporal integration period. Dashed line indicates the mean length of each call type. Colors are: blue—twitter, red—phee, yellow—trill

of mean call length, respectively. Interestingly, these lengths may correspond to timescales of temporal modulations in calls—for twitters, the sum of mean phrase length and mean inter-phrase interval is ~190 ms; for trills, the mean amplitude modulation period is ~30 ms. Thus, as with the initial feature set, MIFs for call classification were also of intermediate length and complexity.

**Accurate classification of novel calls using MIFs alone**. To validate our model and to test the effectiveness of using only the MIFs for classifying call types, we used a novel set of calls consisting of 500 new within-category and 500 new outside-category calls drawn from the same 8 marmosets. This test call set did not significantly differ from the training set along any of the characterized parameters (red histograms in Fig. 1). We conceptualized each MIF as a simulated template-matching neuron whose response to a stimulus was defined as the maximum value of the normalized cross-correlation (NCC) function. This simulated MIF-selective neuron 'spiked' whenever its response crossed its optimal threshold, i.e., when an MIF was detected in the stimulus. In Fig. 5, we plot the spike rasters of simulated MIF-selective neurons for twitter, phee, and trill (top 10 MIFs shown), responding to a train of randomly selected calls from the novel test set. Each spike was weighted by the log-likelihood ratio of the MIF and the weighted sum of responses in 50 ms time bins was taken as the evidence in support of the presence of a particular call type. Although occasional false positives and misses occurred, over the set of MIFs, the evidence in support of the correct call type was almost always the highest. Therefore, production-invariant call categorization is a two-step process—first, MIFs are detected in the stimuli, and then each feature is weighted by its log-likelihood ratio to provide evidence for a call type.

We quantified the performance of the entire set of MIFs ($n =$ 11, 16, and 20 for twitter, phee, and trill, respectively) for the classification of novel calls by parametrically varying an overall evidence threshold and computing the hit rate (true positives) and false alarm rate (false positives) at each threshold. From these data, we plotted receiver operating characteristic (ROC) curves (Fig. 6a). In these plots, the diagonal corresponds to chance, and perfect performance corresponds to the upper left corner. The MIFs achieved >95% detection rates for all call types with very low false alarm rates.

**Control simulations**. First, we ensured that our selection of 6000 initial random features adequately sampled stimulus space. To do so, we iteratively selected sets of MIFs using our greedy search algorithm from initial random sets from which previously picked MIFs were excluded. We found that distinct sets of MIFs that had similar classification performance could be selected in successive iterations (Supplementary Fig. 3). This suggests that our initial random feature set indeed contained several redundant MIF-like features, confirming the adequacy of our initial sampling.

Second, in order to determine the contributions of various model assumptions and parameters, we repeated this process of random initial feature generation, threshold optimization, and MIF selection in different scenarios. To better visualize these differences, we used detection-error tradeoff curves (Fig. 6b), where perfect performance is the lower left corner. In this figure, the performance of the default model, as described above, is plotted in blue. First, when we used the acoustic waveform of calls instead of cochleagrams, classification performance was on average worse (Fig. 6b; red), suggesting that phase information in the waveform may be detrimental for classification. Second, we used the features with top merits without greedy-search optimization for classification, and again found that performance compared to the default model was worse (Fig. 6b, green). Finally,

perfect classification. For twitters, detecting a single feature (the top MIF) was sufficient to gain 0.95 bits of information. Subsequent features probably detected only a few additional twitters without introducing new false alarms. For the other call types, however, the top MIF only provided 0.78 or 0.6 bits of information. Although successive MIFs individually had high merit (second column), they added little information to the top MIF (first column), likely because of redundancy—each MIF could only add a small number of additional hits without introducing new false alarms. However, detecting these features was crucial for solving the task, as they ultimately elevated the total information to >0.9 bits. The MIFs have positive weights, suggesting that they are informative by virtue of their presence (rather than absence) in the target category. Because, we approach very high levels of classification using our pairwise optimization of mutual information, and because joint optimization of mutual information across the entire MIF set is computationally expensive, we used the pairwise-optimized MIF set for all further analyses.

In frequency, MIFs neither encompassed the entire call bandwidth, nor consisted of only few frequency bands. In time, MIFs showed integration windows of the order of hundreds of milliseconds (Fig. 4a–c). The mean MIF lengths were 215 ms, 68 ms, and 406 ms for twitters, trills, and phees, respectively. Compared to the average lengths of the calls (twitters: 1.25 s, trills: 0.5 s, phees: 1.27 s), these correspond to 17%, 14%, and 32%

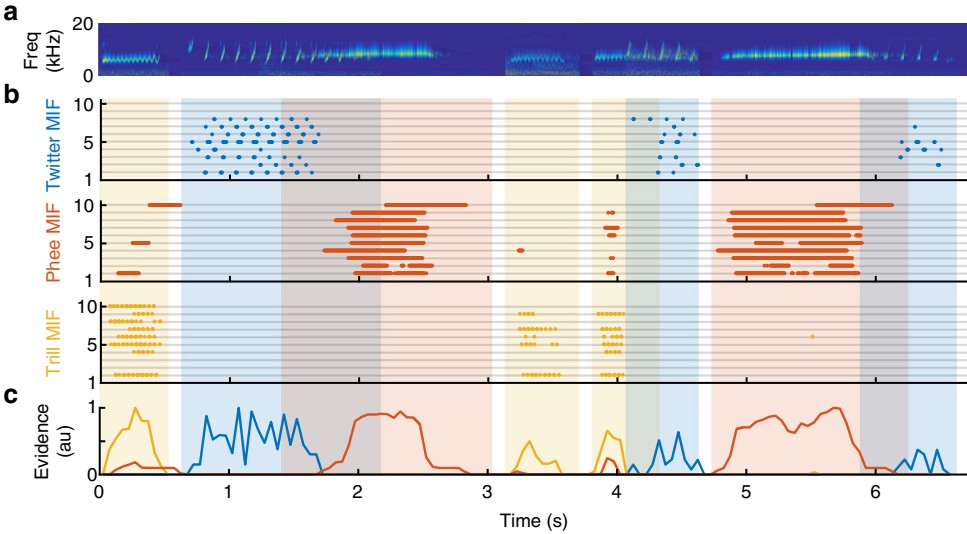

**Fig. 5** MIF responses to marmoset call sequences. **a** The cochleagram of a sequence of marmoset calls, some of which overlap. **b** Raster plot of the responses of the top 10 MIFs for twitter (top, blue), phee (middle, red), and trill (bottom, yellow). Each dot represents spiking of a putative MIF-selective neuron (i.e. when the response of the MIF exceeds its optimal threshold). **c** The evidence for presence of a particular call type, defined as the normalized sum of the firing rate of all MIF-selective neurons, weighted by their log-likelihood ratio. Over the duration of each call, the call type with the most evidence is considered to be present. Occasional false alarms are usually outweighed by true-positive MIF detections

using entire calls as features, either treating entire individual calls as features ('grandmother cell' model; Fig. 6b, yellow), or using the aligned and averaged training call as a single feature (Supplementary Fig. 4) also resulted in worse performance compared to the intermediate feature-based model.

In Fig. 6c, we compare the average cumulative information added by successive features across all three call classification tasks (twitter vs. all other calls, trill vs. all other calls, and phee vs. all other calls) for each control simulation against the performance of the default model. The default model significantly outperformed (at $p < 0.01$, rank-sum test) the no greedy-search model for all classification tasks, after correcting for multiple comparisons (Bonferroni correction). Exact $p$-values corresponding to default model comparison with the constrained model and the no-greedy-search model were: twitter ($p = 0.000087$ and $p = 0.00021$, respectively, rank-sum tests), trill ($p = 0.0058$ and $p = 0.00067$, respectively, rank-sum tests), and phee ($p = 0.00015$ and $p = 0.00021$, respectively, rank-sum tests). While the default model for trill exhibited significantly higher performance compared to the acoustic-waveform model ($p = 0.000091$, rank-sum test), the default models for twitter and phee did not ($p = 0.89$ and $p = 0.43$, respectively, rank-sum tests). These results suggest that our underlying assumptions—using the cochleagram, unconstrained initial feature selection, and MIF optimization using a greedy search—were justified. Twitter MIFs were not qualitatively different when derived from calls emitted by a smaller set of animals (4 animals). Training on a set of 4 animals and testing on the other 4 animals yielded high performance (Fig. 6d, triangles), confirming the robustness of using MIFs for categorization of new calls. Twitter MIF performance in classifying twitters from other twitters was near-chance, suggesting that the estimation of mutual information values was unbiased (Fig. 6d, circles). Finally, MIFs derived for one task (such as trill vs. other calls) showed chance level performance for other tasks (such as twitter vs. other calls; Fig. 6d, crosses), demonstrating the task-dependence of the derived MIFs.

**The precedence of intermediate features for classification.** We have previously shown that features of intermediate lengths and complexities possess high individual merits for classification (Supplementary Fig. 2). We have also shown that the set of MIFs is composed intermediate features (Fig. 4a–c). To directly test whether features of intermediate size were indeed the most informative, we re-derived MIFs after constraining the initial set of features to particular time and frequency bins and quantified model performance (Fig. 7). When we constrained the features to be only small (<100 ms and <1 oct.) or removed all small features, performance was worse than the default model (Fig. 7, top row). Similarly, model performance was worse compared to the default model when we constrained to only large features (>250 ms and >2 oct.), or removed all large features. When we constrained bandwidth and time independently to be large or small, model performance was worse compared to the default model, with large values being more detrimental (Fig. 7, bottom row). As previously discussed, using the largest possible features (whole calls or average call) resulted in poor classification performance as well. These results demonstrate that features of intermediate size indeed provide the best classification performance.

**MIF tuning properties match neural responses from A1 L2/3.** So far, we have demonstrated MIFs derived purely using theoretical principles can achieve high levels of production-invariant call categorization performance. We then asked whether the auditory system uses such an optimal feature-based approach for call classification. To explore this possibility, as a first step, we generated tuning curves of model neurons that were selective for the theoretically derived MIFs, and asked if these tuning curves matched previous experimental observations. In this effort, we were restricted by the appropriateness and availability of previous data. To do so, we first constructed cochleagrams of stimuli, such as trains of frequency-modulated sweeps, amplitude modulated tones, noise bursts, clicks, two-tone combinations, etc. We then used the maximum value of the NCC function as a metric of the model MIF neurons' response to these stimuli, as we did earlier for calls. These responses were conceptualized as membrane potential responses, which elicited spiking only if they crossed each MIF neuron's optimal threshold. We used a power law nonlinearity, applied to the maximum NCC values (see Methods,

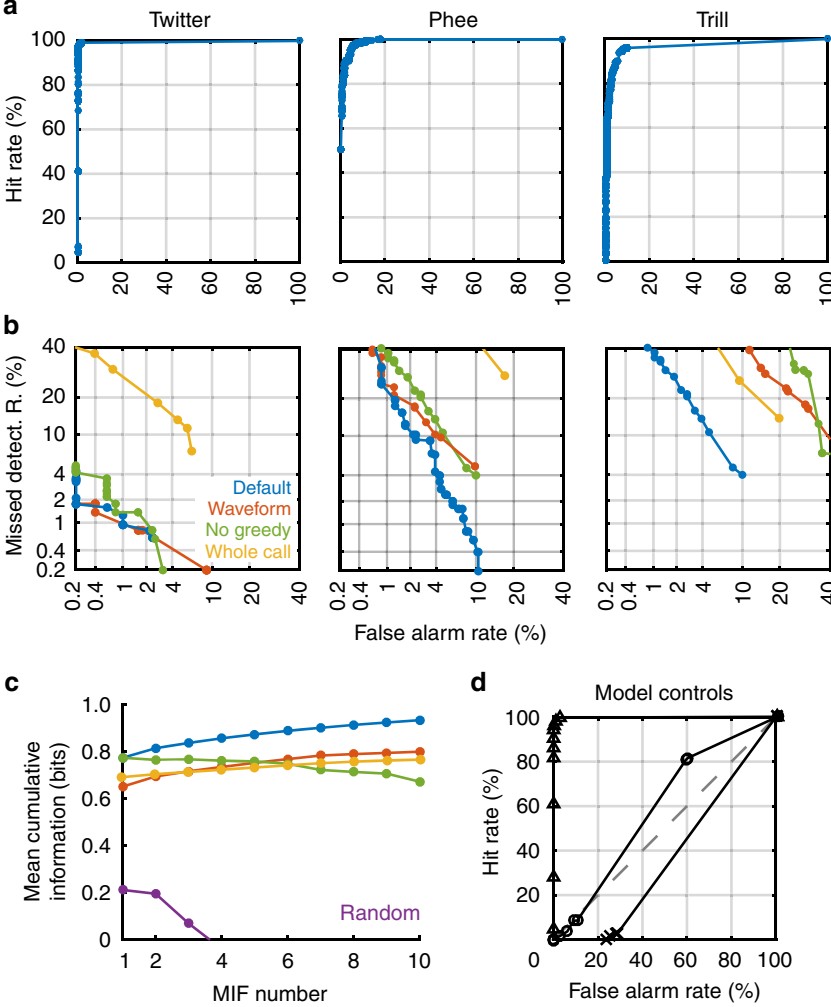

**Fig. 6** Classification performance and controls. **a** Receiver operating characteristic (ROC) curves for the classification of twitters, phees, and trills using MIFs alone. **b** Detection error tradeoff (DET) curves comparing the default model (blue) to other model variations: (i) MIF-based classification with acoustic waveforms (red), (ii) feature selection without greedy search (green), and (iii) when entire calls are used as features (yellow). **c** Comparison between various model conditions (same as B) in terms of cumulative information added by each successive feature, averaged across all three call type classification tasks. Random (purple) is the classification of twitters using randomly selected features as MIFs, averaged across 20 trials. **d** ROC curves of three model controls: (i) classification of twitters when the model is trained on the twitters of 4 animals, and tested on twitters from 4 new animals (triangles), (ii) classification of twitters from other twitter calls (circles), and (iii) classification of twitters using trill MIFs (crosses)

Equation 2), to determine the firing rate responses of model MIF neurons (Supplementary Fig. 6). We then compared these model MIF tuning curves to neural data from marmoset primary auditory cortex (A1).

Although, the MIF model did not have prior access to neurophysiological data, we found that model MIF neural tuning recapitulated actual data to a remarkable degree, both at the population and single-unit levels. For example, the population of model MIFs showed high preference for natural calls compared to reversed calls (Fig. 8a, bottom), similar to observations by Wang and Kadia[18] (reproduced in Fig. 8a, top). The high sparseness of auditory cortical neurons is well-documented[19–21]. The responses of model MIF-selective neurons were also sparse—only few MIF neurons were activated by any given stimulus set, and only after extensively optimizing the parameters of the stimulus set to drive-specific model MIF neurons. For example, in Fig. 8b (top), we show a single-unit recording from a marmoset A1 L2/3 neuron that did not respond to most stimulus types (reproduced from Sadagopan and Wang[21]), and only strongly responded to two-tone stimuli. Twitter MIFs (Fig. 8b, bottom) were similarly not

responsive to most stimulus types, and only responded to carefully optimized linear frequency-modulated (lFM) sweeps. None of the model twitter and trill MIF-selective neurons responded to pure tones (Fig. 8b, bottom), similar to many A1 L2/3 neurons.

Most strikingly, we could recapitulate some specific and highly nonlinear single-neuron tuning properties as well. Figure 8c (top; reproduced from Sadagopan and Wang[21]) is a single-unit recording from marmoset A1 L2/3 that did not respond to pure tones, but selectively responded to upward lFM sweeps of specific lengths (~80 ms). Responses of at least three of the top 5 twitter MIF-selective model neurons showed similar tuning for 80 ms-long upward lFM sweeps (Fig. 8c, bottom). A second peak at ~40 ms was also present in responses of two model twitter MIF-selective neurons, also matching the experimental data. Figure 8d (top; reproduced from Sadagopan and Wang[21]) shows another single-unit recording from marmoset A1 L2/3, where the neuron did not respond to single lFM sweeps (lightest gray line), but strongly responded to trains of upward lFM sweeps occurring with 50 ms inter-sweep interval. The neuron's response scaled

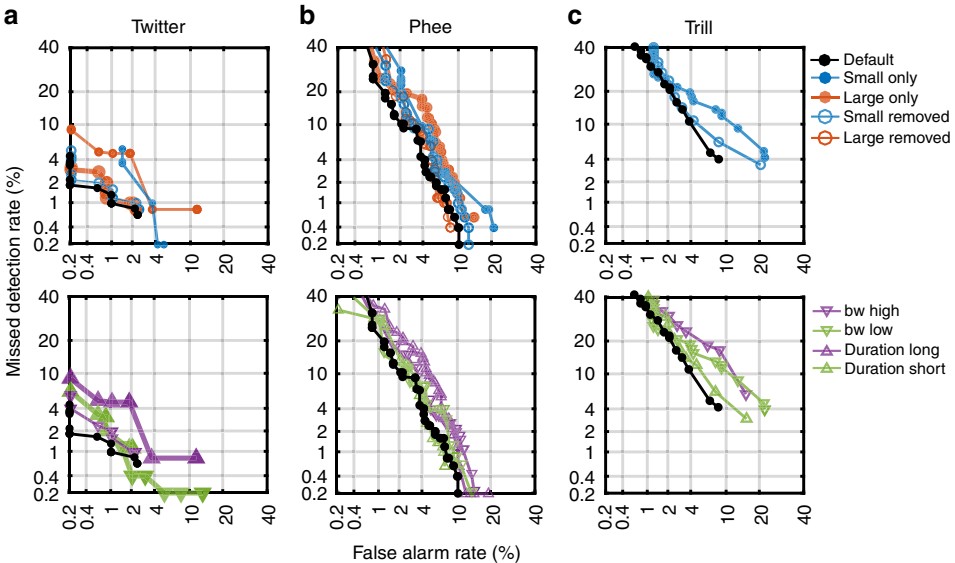

**Fig. 7** The precedence of intermediate features for classification. DET curves for call classification using features of different sizes, bandwidths, and durations for the classification of **a** twitters, **b** phees, and **c** trills. The default model is in black. Top row shows performance when using small features only (<100 ms and <1 oct; blue discs.) or excluding small features (blue circles), and using large features only (>250 ms and >2 oct.; red discs) or excluding large features (red circles). For trills, some of these conditions fall outside the range of the axes. Bottom row shows performance when feature bandwidths and durations were independently varied. Because of the short duration of trill calls, we did not test the effect of using only long duration features. Symbols are: purple inverted triangles—high bandwidth features only, green inverted triangle—low bandwidth features only, purple triangle—long-duration features only, green triangle—short-duration features only

with the number of sweeps present in the train (darker colors correspond to more sweeps). Three of the top 5 twitter MIF-selective neurons also showed remarkably similar tuning (Fig. 8d, bottom)—these model neurons did not respond to single sweeps, but responded to trains of at least 2 or more sweeps occurring with a 50 ms inter-sweep interval. Taken together, these data suggest neurons tuned to MIF-like features are present in A1 L2/3. Therefore, we predict that a spectral-content based representation of calls in the ascending auditory pathway becomes largely a feature-based representation in A1 L2/3.

Consistent with the prediction of feature selectivity, we also found neurons in A1 of both marmosets and guinea pigs that respond selectively to conspecific call features. In Fig. 9, we present the spike rasters of example single neurons in both marmoset and guinea pig A1 responding to marmoset (Fig. 9a) and guinea pig calls (Fig. 9b), respectively. We presented multiple exemplars of each call type as stimuli. These example neurons responded at specific time points to a few call stimuli, typically across 1–3 categories. Such responses are consistent with our feature-based model because single features alone do not completely categorize calls, i.e., MIFs do not have 1 bit of information for categorization. Rather, combinations of features weighted by their log-likelihood ratios are necessary to ultimately achieve complete call category information. These data provide promising support for our model, but further experiments are necessary to: (1) determine how informative these neural features are about call category and how they compare with model features, (2) to confirm where such responses arise in the auditory pathway, and (3) to account for possible low-level confounds. Experiments are presently ongoing to address these issues.

**Task-dependent MIF detection as a general computation**. To determine whether MIF-based representations of sounds could also be used for optimally solving other tasks, we performed three proof-of-principle simulations using limited available datasets. First, we tested whether we could accurately determine caller

identity using an MIF-based approach. We generated training and test sets of 60 twitters each from eight marmosets, and generated 500 initial random features from the training set. We applied the greedy-search algorithm to determine the MIFs for caller identification in a caller A vs. all other callers task (Fig. 10a). We found that similar to call categorization, caller identification could also be achieved using a small number of MIFs (n = 4). If caller identification was performed in a binary fashion (four classifications between two animals each), in half of these tasks, classification could be accomplished using less than 3 MIFs, indicating that the calls of these marmosets probably differed along the frequency axis. This is because if there are clear differences in dominant frequency (for example, Animal 1 vs. 4 in Fig. 1d), all features that lie in one animal's frequency range will detect all of that animal's calls and none of the other animal's calls. During the greedy search procedure, these features will be considered redundant and reduced to a single feature. In the other half, more MIFs were required for caller identification, and in general, MIFs were larger than those for call-type classification. This is likely because the differences between twitters produced by these animals are smaller compared to the differences between call types and can only be resolved in a higher dimensional space. Thus, integration over more frequencies and a larger time window may be necessary to resolve caller differences. In Supplementary Fig. 7, we plot the ROC for caller identification between a pair of marmosets with overlapping dominant frequencies. The MIF-based approach (n = 20 MIFs) achieved >80% hit rates with <10% false alarm rate for caller identification.

Second, we tested whether MIF-based call classification generalized to other vocal species, using guinea pig and macaque call classification as examples. Guinea pigs are highly vocal rodents that produce seven primary call types[22–24], which are highly overlapping in the low frequency end of the spectrum, and show high production variability. We used the MIF-based approach to classify guinea pig call types (whine, wheek, and rumble) from all other guinea pig call types. Similar to marmosets, guinea pig classification could be accomplished using

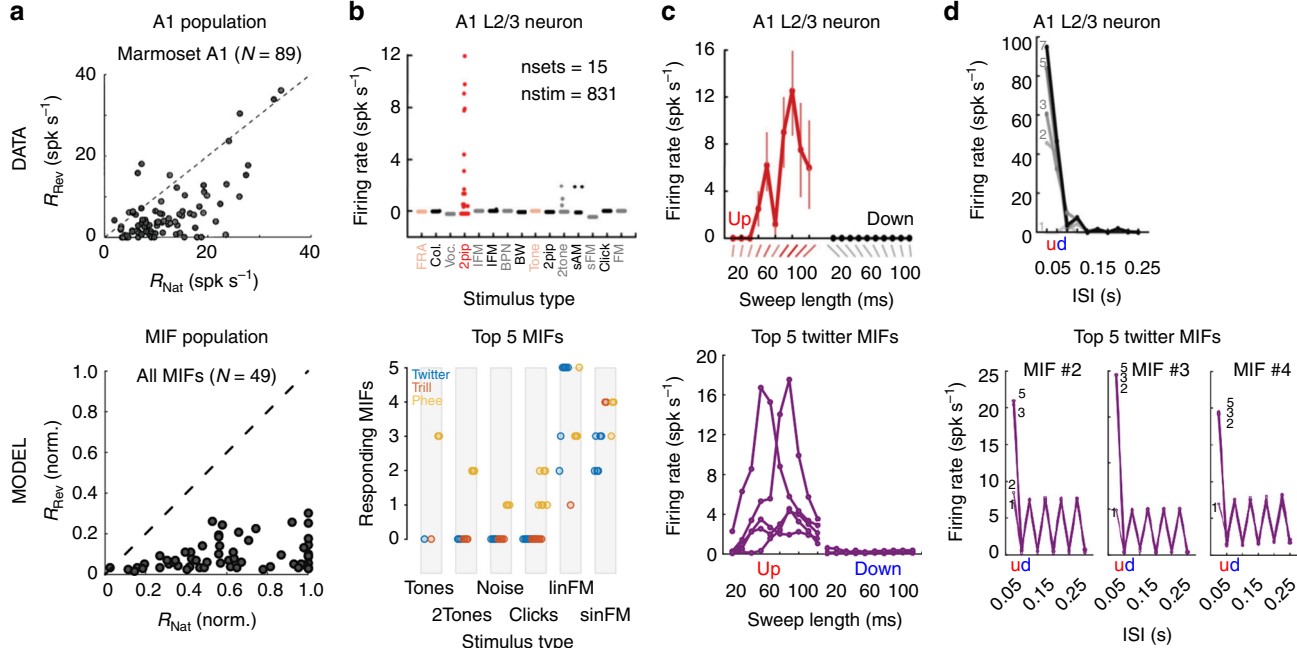

**Fig. 8** Predictions of putative MIF-neuron tuning properties match cortical data. (**a–d**, top row) Neural data from marmoset A1. (**a–d**, bottom row) Model predictions. (**a**-top) Preference of marmoset A1 responses for natural twitters over time-reversed twitters. (**a**-bottom) Preference of model MIF neurons for natural calls over reversed calls. (**b**-top) Sparse responses of marmoset A1 L2/3 neuron. (**b**-bottom) Sparse responses of MIF neurons. The number of MIF neurons showing responses to the stimulus categories on the x-axis are plotted. Colors correspond to call type (blue—twitter, red—trill, yellow—phee). (**c**-top) Marmoset A1 L2/3 neuron tuned to upward lFM sweeps of a specific length (~80 ms). Error bars correspond to ±1 SD. (**c**-bottom) Twitter MIF neurons show similar tuning. (**d**-top) Marmoset A1 L2/3 neuron that does not respond to single lFM sweeps but shows tuning to trains of upward lFM sweeps with 50 ms inter-sweep interval. Grayscale corresponds to the number of lFM sweeps in the train. (**d**-bottom) Three of the top 5 twitter MIFs showed similar tuning for lFM sweep trains. **a**-top reproduced from Wang and Kadia (2001), **b–d** top reproduced from Sadagopan and Wang (2009)

a handful of features (12, 9, and 3 MIFs for whine, wheek, and rumble), and MIF-based classification achieved high performance levels (Fig. 10b). Similarly, we implemented the MIF-based algorithm to classify macaque calls (using 5, 4, and 9 MIFs for coos, grunts, and harmonic arches) from a limited macaque call dataset[25] and achieved high classification performance (Fig. 10c). These proof-of-principle experiments demonstrate that an MIF-based approach indeed succeeds for different auditory classification tasks and in different species, suggesting that building representations of sounds using task-relevant features in auditory cortex may be a general auditory computation.

## Discussion

In these experiments, we set out to understand the computations performed by the auditory system that enable the categorization of behaviorally critical sounds, such as calls, despite wide variations in the spectrotemporal structure of calls belonging to a category (production variability). We found that the optimal theoretical solution is to detect the presence of informative mid-level features (termed MIFs) in calls. These MIFs generalize over production variability, and conjunctions of MIFs accomplish production-invariant call classification with high accuracy. Critically, the tuning properties of model MIF-selective neurons matched previous recordings from marmoset A1 to a surprising degree. MIF-based classification was also successful for other tasks (marmoset caller identification), and in other species (guinea pig and macaque call recognition). Our results suggest that the representation of sounds in higher auditory cortical areas is based on the detection of optimal task-relevant features.

An implication of our results is that in higher auditory processing stages, neural representations of sounds serve specific

behavioral purposes. For example, the MIF-based classification approach that we proposed here is targeted to solve well-defined classification problems. At earlier stages of the auditory pathway, however, it may be more important to faithfully represent sounds using basis sets that enable the accurate and complete encoding of novel stimuli. Previous theoretical studies have proposed, for example, that natural sounds can be efficiently encoded using spike patterns, where each spike represents the magnitude and timing of input acoustic features[26]. However, when optimized to encode the complete waveforms of natural sound ensembles, the kernel functions that elicit each spike show a striking similarity to cochlear filters. The advantage of this approach is that novel stimuli can be completely encoded using these kernel functions. In our approach, the input to our model implements a similar encoding schematic—in the cochleagram, inputs are encoded as spatiotemporal spike patterns, where each spike is the result of cochlear filtering. In this early representation, while information about category identity is present, it is distributed in the activity of many neurons in a high-dimensional space. We propose that in later processing stages, this early representation is transformed into a representation where category identity is more easily separable. By encoding MIF-like features, sound representation in later processing stages is less useful for high-fidelity encoding (although stimulus reconstruction is possible, see Supplementary Note and Supplementary Fig. 5), but is instead goal-oriented. However, this means that each task will require a distinct set of MIFs for optimal performance, and animals likely perform a large number of such behaviorally relevant tasks. The observed >1000-fold increase between the number of cochlear inputs and auditory cortical neurons may partially result from this necessity to encode a multitude of task-dependent MIFs. Previous theoretical studies have suggested that the generation of redundant and over-

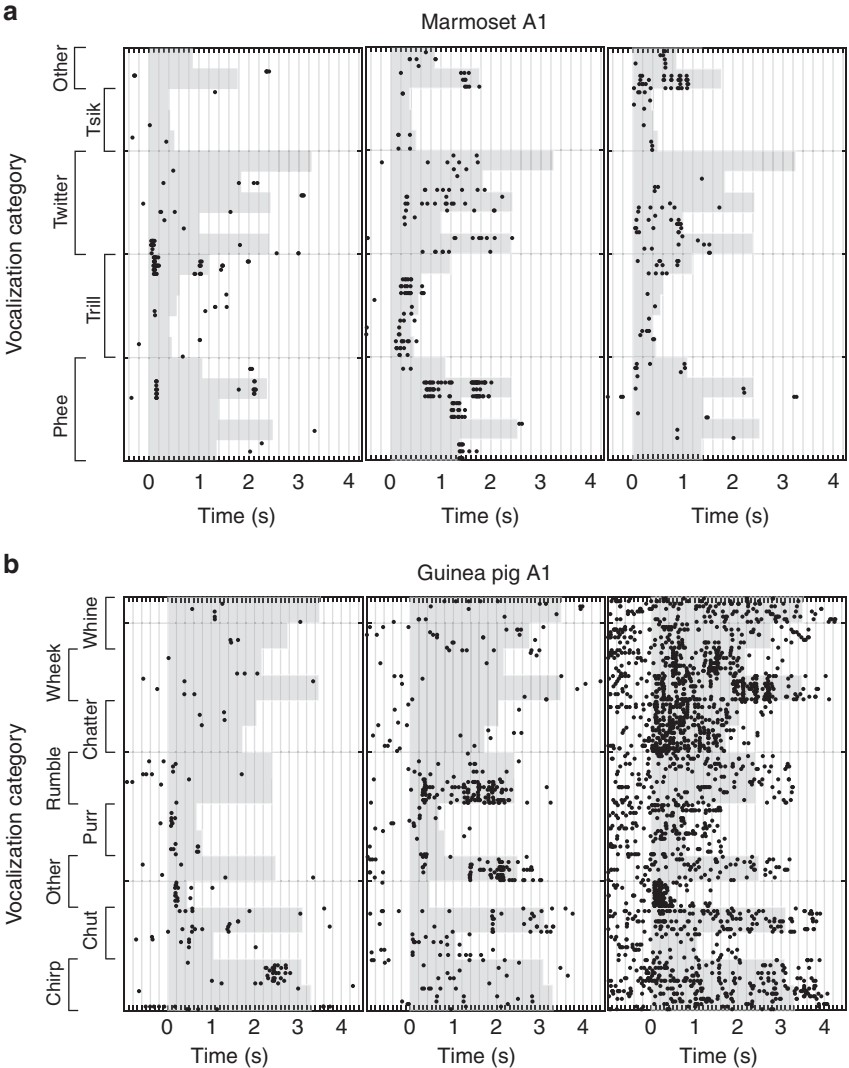

**Fig. 9** Feature selectivity in cortical neurons. **a** Spike rasters of three single units from marmoset A1 responding to marmoset call stimuli. Black dots correspond to spikes; gray shading corresponds to stimulus duration (different calls have different lengths). Note that spikes occur at specific times, and in response to 2 or 3 call types, suggesting that the neurons are responding to smaller features within these calls. **b** Spike rasters of three single units from guinea pig A1 responding to guinea pig call stimuli

complete representations of sounds to solve spatial localization problems might underlie this increase in the number of neurons[27]. Our study proposes another computational reason why such an expanded representation of sounds may be necessary.

Another powerful method to accomplish classification uses hierarchical convolutional neural networks, or deep networks. In these models, layers of filtering, normalization, and pooling operations are cascaded, resulting in individual units exhibiting increasingly complex tuning properties[28–30]. A final layer reads out class identity. Deep networks can achieve near-human levels of performance on specific tasks, but carry some disadvantages. First, they often require training data of the order of millions of samples. In the visual domain, deep networks appear not to use the same features as humans for object classification[31]. Finally, an intuitive explanation for how deep network models actually accomplish classification is not yet available. In our approach, we explicitly train our MIF neurons to extract maximally distinguishing features, providing insight into why certain features are represented amongst these neurons. Our model does not require as extensive a training set. We consider our approach

complementary to the deep learning approach, in that we aim to provide an explicit and intuitive explanation of why certain features are extracted, as opposed to matching human performance using complex model architectures.

Conceptually, our MIFs may be similar to 'image signatures' obtained by recently developed unsupervised methods[32] (see Supplementary Discussion). Our approach is complementary to alternative experimental approaches, such the characterization of neural tuning along an exhaustive list of call parameters[33], characterizing call tuning as tuning for regions of the modulation spectrum[34–36], and combinations of these methods in conjunction with machine learning tools[37] (see Supplementary Discussion). Our results suggesting auditory cortex as a locus where the neural representation of vocalization sounds generalizes over production variability is consistent with a recent study showing that neurons in the auditory cortex of ferrets show robust responses to vowel identity tolerant to manipulations of various vowel features[38].

Mechanistically, neural selectivity for MIFs may be generated (1) gradually along the ascending auditory pathway, or (2) de

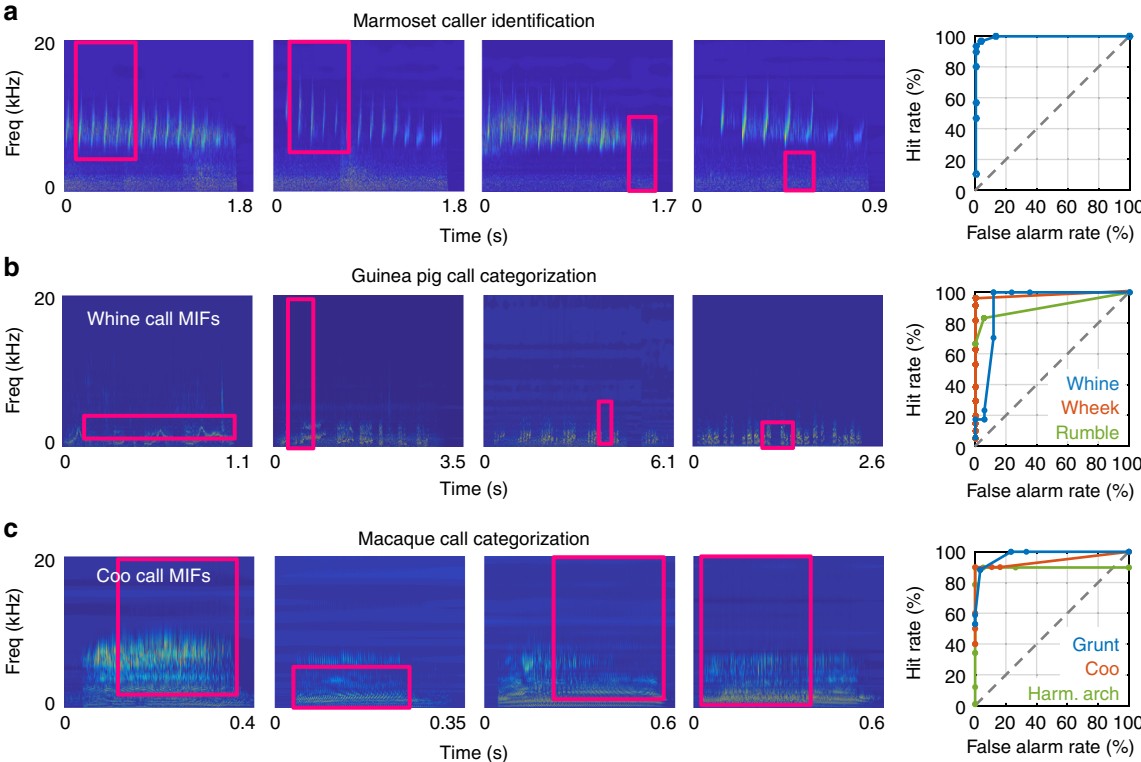

**Fig. 10** The applicability of MIF-based classification for other auditory tasks. The top four MIFs and ROC curves for: **a** marmoset caller identification (twitter calls), **b** Guinea pig call classification (MIFs for whine calls shown). Colors are: blue—whine, red—wheek, green—rumble. **c** Macaque call classification (MIFs for coo calls shown). Colors are blue—grunt, red—coo, green—harmonic arch

novo in cortex. Single-neuron feature selectivity often (but not always, see below) leads to selectivity for one or a few call types, and analyzing call selectivity of neurons at different auditory processing stages could provide insight into where MIF-based representations might be generated in the auditory pathway. In early auditory processing stages, evidence for call selectivity at the single-neuron level is minimal. For example, at the level of the cochlear nucleus, few single neurons in species other than mice show call selectivity[39]. At the level of inferior colliculus, a population-level bias in call-selectivity has been reported[39–41], but evidence for single-neuron level call-selectivity is equivocal[42]. It is only at the level of auditory cortex where clear single-neuron selectivity for calls or call features has been observed. Therefore, it is quite likely that selectivity for MIF-like features in species with spectrotemporally complex calls is generated at the level of auditory cortex. This is supported by the expansion in the number of cortical neurons mentioned above. Importantly, the cortical emergence of MIF-based representations is also supported by the fact that MIF-like responses have been observed in the superficial layers of marmoset A1[21].

We propose the following hierarchical model for auditory processing based on the representation of task-relevant features. In thalamorecipient layers of A1, representation of sound identity is still based on spectral content. This is reflected in the strongly tone-tuned responses of A1 L4 neurons. From these neurons, tuning for MIF-like features may be generated using nonlinear mechanisms, such as combination-sensitivity. For example, the tuning properties of the marmoset A1 responses shown in Fig. 8 was determined to be the result of selectivity for precise spectral and temporal combinations of two-tone pips[21]. This is also consistent with a recent computational model showing that combinations of spectrotemporal kernels, optimized for representing natural sounds, recreates aspects of experimentally observed spectrotemporal receptive fields from recordings in cat auditory cortex[43]. Further experiments, probing call and feature selectivity in identified layers of A1, are necessary to more precisely address where selectivity for MIF-like features first emerges in the ascending auditory pathway, and at what stage MIFs are combined to result in a categorical read-out. Once categories are detected, further hierarchical processing stages might be necessary to accomplish more sophisticated behavioral goals, such as caller identification, integration of social context with call perception, or decoding the emotional valence of calls.

In conclusion, we propose a hierarchical model for solving a central problem in auditory perception—the goal-oriented categorization of sounds that show high within-category variability, such as speech[1,2] or animal calls[3]. Our work has broad implications as to where in the auditory pathway categorization begins to emerge, and what features are optimal to learn in categorization tasks. For example, the lack of distinction of perceptual categories of English /r/ and /l/ by native Japanese speakers might be a consequence of not learning and encoding[44,45] the optimal features necessary for this /r/-/l/ categorization, as it is not task-relevant for Japanese speech. Our model would predict that /r/-/l/ category learning would cause selective responses to develop for new task-relevant features, and primarily reflected in changes to the A1 L2/3 circuit. Consistent with this hypothesis, a recent study showed that training humans to categorize monkey calls resulted in finer tuning for call features in the auditory cortex[46]. We therefore suggest that the neural representation of sounds at higher cortical processing stages uses task-dependent features as building blocks, and that new blocks can be added to this representation to enable novel perceptual requirements.

## Methods

**Vocalizations**. All procedures conformed to the NIH Guide for Care and Use of Laboratory Animals. All marmoset procedures were approved by the Institutional Animal Care and Use Committee (IACUC) of The Johns Hopkins University. All guinea pig procedures were approved by the IACUC of the University of Pittsburgh. We used vocalization recordings from 8 adult marmosets, both male and female, for these experiments. Marmoset calls were recorded from a marmoset colony at The Johns Hopkins University using directional microphones[8]. Guinea pig calls were recorded from 3 male and 3 female adult guinea pigs. Two or more guinea pigs with varied social relationships were placed on either side of a transparent divider in a sound attenuated booth. Directional microphones, suspended above the guinea pigs were used to record calls. Calls were recorded using Sound Analysis Pro 2011[47], digitized at a sampling rate of 48 KHz, low-pass filtered at 24 KHz, manually segmented using Audacity, and classified into different call types.

**Random feature generation**. All modeling was implemented in MATLAB. We focused on classifying each of three major marmoset call types, twitter, trill, and phee, from all other call types. That is, three main binary classification tasks—twitter vs. all other calls, trill vs. all other calls, and phee vs. all other calls were considered. We set up the categorization tasks as a series of binary classifications based on the results of an earlier study of visual categorization that demonstrated the advantages of features learnt using multiple binary classifications compared to those learnt using a single multi-way classification. Specifically, in that study, multiple binary classifications resulted in features that were distinctive and highly tolerant to distortions[48]. For each classification task, we first generated training datasets, which consisted of 500 random within-class calls (e.g., twitters) produced by 8 animals (about 60 calls per animal), and 500 random outside-class calls (e.g., trills, phees, other calls) produced by the same 8 animals. In order to convert sound waveforms of the calls into a physiologically meaningful quantity, we transformed these calls into cochleagrams using a previously published auditory nerve model[49] using human auditory nerve parameters with high spontaneous rate. We used human auditory nerve parameters because of the close similarity between marmoset and human audiograms[50]. The output of this model was the time-varying activity pattern of the entire population of auditory nerve fibers, and resembles the spectrogram of the call (Fig. 2a, b). We then extracted 6000 random features from these 500 within-class cochleagrams. To do so, we randomly chose a center frequency, bandwidth, onset time and length and extracted a snippet of activity from the cochleagram. Each feature thus corresponded to the spatiotemporal pattern of activity of a subset of auditory nerve fibers within a specified time window (magenta box in Fig. 2b). We used rectangular feature shapes rather than other shapes to minimize assumptions – for example, an ellipse shaped feature would imply that the weighting of individual auditory nerve fibers changes over time. For twitters, to ensure that smaller features were well-sampled, 2000 of these features were restricted to have a bandwidth less than 1 octave and a duration less than 100 ms. The bandwidth and duration of the remaining 4000 features were not constrained.

**Feature complexity**. We characterized feature complexity using the reduced kurtosis of the activity distribution of all auditory nerve fibers contained within a feature. Briefly, if the feature was an empty region of the cochleagram, or a region of uniform activity, the activity of all nerve fibers in all time bins would be about equal. This activity would thus be normally distributed, and show a reduced kurtosis value of zero. At the other extreme, for entire calls, there would be many bins of high activity, and a large number of bins with zero activity, resulting in an activity distribution with very high reduced kurtosis. We hypothesized that mid-level features that represent aspects of calls such as frequency-modulated sweeps or combinations of phrases over time would show intermediate reduced kurtosis values, and be more informative than low-level (tones) or high-level (entire calls) features.

**Threshold optimization**. We defined the response of a feature to a call as the maximum value of the normalized cross-correlation (NCC) function between the feature's cochleagram and the call's cochleagram, restricted to the auditory nerve fibers that are represented in the feature. Note that this means features can only be detected in the frequency range that they span, but can be detected anywhere in time within a call. NCC is a commonly used metric to quantify template-match. To compute the NCC, the feature and the cochleagram patch at each lag were normalized by subtracting their respective mean values and dividing by their respective standard deviations before convolving them. This results in a value between −1, signifying that the feature and cochleagram patch at that lag are completely anti-correlated, and +1, signifying a perfect match between the feature and the cochleagram. Because this is a computation-intensive step, template matching was implemented on an NVIDIA GeForce 980 Ti GPU. For each feature, then, we obtained 500 within-class responses, and 500 outside-class responses (response histograms of an example feature in Fig. 2c). To transform these continuous response distributions into a binary detection variable, we used mutual information to quantify the information provided by a feature about the class (within- or outside-class) over a parametrically varied range of thresholds. We computed mutual information following the method of Ullman et al.[12], by measuring the

frequency of detecting a feature $f_i$ at a given threshold $\theta_i$ ($f_i = 1$ if present, 0 if absent) in the within-class ($C = 1$) or outside-class ($C = 0$) cochleagrams as:

$$I(f_i(\theta_i), C) = \sum_{\substack{f_i=\{0,1\}\\C=\{0,1\}}} p(f_i, C) \log\left(\frac{p(f_i, C)}{p(f_i)p(C)}\right) \tag{1}$$

where $p(C)$ was assumed to be 0.10. We empirically verified that features identified were insensitive to variations of this value. The optimal threshold for each feature was taken to be the threshold value at which the mutual information was maximal, and the merit of each feature was taken to be the maximum mutual information value in bits (Fig. 2c). The weight of each feature was taken to be its log-likelihood ratio. At the end of this procedure, each of the initial 6000 features were allocated a merit, a weight, and an optimal threshold at which each individual feature's utility for classifying calls as belonging to within- or outside-class was maximized. Note that merit and weight are distinct quantities that need not be monotonically related. For example, if the lack of energy in a frequency band is indicative of a target category, features that contain energy in this frequency band will be detected often in the other categories, but not in the target category. The feature will thus have high merit for classification, as it is informative by its absence, but have a negative weight.

**Greedy search**. Because we chose initial features at random, many of these features individually provided low information about call category, and many of the best features for classification were similar, or redundant. Therefore, to extract maximal information from a minimal set of features for classification, we used a greedy search algorithm[12] to iteratively (1) eliminate redundant features, and (2) pick features that add the most information to the set of selected features. The minimal set of features that together maximize information about call type were termed maximally informative features (MIFs). The first MIF was chosen to be the feature with maximal merit from the set of all 6000 initial random features. Every consecutive MIF was chosen to maximize pairwise added information with respect to the previously chosen MIFs. Note that these consecutive features need not have high merit individually. We iteratively added MIFs until we could no longer increase the hit rate without increasing the false alarm rate. Practically, this meant adding features until total information reached 0.999 bits, or individual features added less than 0.001 bits, whichever was reached earlier. At the end of this procedure, a small set of MIFs, containing the optimal set of features for call classification was obtained.

**Analysis and statistics**. To test how well novel calls could be classified using these MIFs alone, we generated from the same 8 animals a test set of 500 within- and outside-class calls that the model had not been exposed to before. We computed the NCC between each test call and MIF, and considered the MIF to be detected in the call if the maximum value of the NCC function exceeded its optimal threshold. If detected, the MIF provided evidence in favor of a test call belonging to a call type, proportional to its log-likelihood ratio. We then summed the evidence provided by all MIFs and generated ROC curves of classification performance by systematically varying an overall evidence threshold. We used the area under the curve (AUC) to compare ROC curves for classification performance by MIFs generated with different constraints (see Results). Statistical significance was evaluated using rank-sum tests, with Bonferroni multiple-comparisons corrections, for comparing between these conditions, and for comparing performance to a large number of simulations generated using random MIFs.

**Generating predictions**. To generate predictions of the responses of putative MIF-selective neurons to other auditory stimuli, we first generated a large battery of stimuli that have been used in previous recordings from marmoset A1, and computed their cochleagrams as earlier. We then computed the maximum value of the NCC function between the MIF and the stimulus cochleagram. This resulted in response values that could be conceptualized as equivalent to membrane potential ($V_m$) responses. These were converted to firing rates by applying a power law nonlinearity, of the form:

$$FR = k \cdot \lfloor V_m - \theta \rfloor^p, \tag{2}$$

where FR is the firing rate response in spk s$^{-1}$, $\theta$ is the MIF's optimal threshold, $p$ is the exponential nonlinearity set to a value of 4, and $k$ is an arbitrary scaling factor.

**Call reconstruction from MIFs**. To reconstruct calls, we conceptualized MIFs as MIF-selective neurons, and considered the times at which NCC values exceeded the optimal threshold to be the spike times of these neurons. MIF spike times were computed with a time resolution of 2 ms to simulate refractoriness, and alpha-functions were convolved with the spike times to determine the peak time at which each MIF was detected. A copy of the MIF cochleagram was then placed at the peak time, or summed (with log-likelihood weights) if overlapping with a previously placed cochleagram. The accuracy of reconstruction was defined as the NCC between the original stimulus and its reconstructed version at zero lag.

**Electrophysiology methods**. Predictions generated from the MIFs were compared to earlier recordings from marmoset A1. All recordings were from the auditory cortex of adult marmosets. Population data comparing natural to reversed twitters were obtained from Wang and Kadia[18]. These experiments were performed in anesthetized marmosets. Single-neuron data regarding feature selectivity were obtained from Sadagopan and Wang[21]. These recordings were from awake, passively listening marmosets. Single-neuron data regarding feature selectivity in guinea pigs were obtained from adult, head-fixed, passively listening guinea pigs at the University of Pittsburgh. Briefly, a headpost and recording chambers were secured to the skull using dental cement following aseptic procedures. Animals were placed in a double-walled, anechoic, sound attenuated booth. A small craniotomy was performed over auditory cortex. High-impedance tungsten electrodes (3–5 MΩ, A-M Systems Inc. or FHC, Inc.) were advanced through the dura into cortex to record neural activity. Stimuli were generated in MATLAB, converted to analog (National Instruments), attenuated, power-amplified (TDT Inc.), and presented from the best location in an azimuthal speaker array (TangBand 4" full-range driver). Single units were sorted online using a template matching algorithm (Ripple, Inc), and refined offline (MKSort). All analyses were performed using custom MATLAB code.

**Reporting summary**. Further information on experimental design is available in the Nature Research Reporting Summary linked to this article.

## Code availability

Custom code will be provided upon request to the corresponding author (S.S.).

## Data availability

The data available upon request from the authors. Some data are from sources for which requests should be made to the original authors.

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

## Acknowledgements

We thank Dr. Yale E. Cohen for generously providing us with a set of macaque vocalizations. We thank Dr. Karl Kandler for many helpful discussions and advice. Patrick Haggerty and Vighnesh Viswanathan contributed to some of the analyses. Samuel Li and Isha Kumbam assisted in the recording and curation of guinea pig vocalizations. S.S. was supported with funding from the NIH (NIDCD R01DC017141), the Departments of Otolaryngology and Neurobiology, the Pennsylvania Lions Hearing Research Foundation, and the Samuel and Emma Winters foundation. X.W. was supported by NIDCD R01DC003180. S.T.L. was supported by a T-32 Training grant in Auditory and Vestibular Neuroscience (NIDCD T32DC011499).

## Author contributions

S.S. designed and implemented an initial version of the model with advice and vocalization data provided by X.W. S.T.L. and S.S. were responsible for all subsequent model development and comparisons of the model to data. P.M.L. collected and analyzed neural data from guinea pig auditory cortex, and recorded and categorized guinea pig vocalizations. S.T.L. and S.S. co-wrote the manuscript with inputs from X.W.

## Additional information

**Competing interests:** The authors declare no competing interests.

