## [Peer Review File · Nature Communications]

Reviewers' Comments:

Reviewer #1:

Remarks to the Author:

This paper clearly demonstrates that production-invariant classification of calls can be achieved using mid-level acoustic features. This is an important result and would be of interest to a wide audience.

The paper was a joy to read. The results and the analyses were presented convincingly. I applaud the authors for the clarity of their writing.

My major objection is to the title of the paper. Auditory recognition is much broader than call classification, which is the main focus of the paper, and caller identification. I suggest to change the title to one that reflects the scope of the paper.

Other minor comments are:

1) The auditory nerve model that transforms calls into cochleograms use human auditory nerve parameters. Please comment on the applicability of these model to marmosets.

2) Please provide more details to how you estimate mutual information.

3) Color bar is missing in Supp. Fig. 1

4) Figure 1C is hard to interpret.

5) Could the authors comment on the relationship of their work to "Unsupervised learning of invariant representations" (doi: 10.1016/j.tcs.2015.06.048). This work also uses random features extracted from stimuli to build invariant representations, however in an unsupervised way.

Reviewer #2:

Remarks to the Author:

Major concerns

1. The methodology of extracting meaningful "features" and the resulting high classification accuracy look impressive at first glance. But on closer inspection it appears to boil down to "if we select those parts of a call representation that distinguish very well between the calls, and then use them to distinguish between the calls, they distinguish very well". Maybe I am missing a crucial point here, but then the authors should probably make this point clearer.

2. The authors emphasize that the classification is based on "mid-level features" but they never state clearly what they consider "mid-level" features, and, by extension, what would be "low-level" or "high-level" features. It might be that the authors simply mean "cochleogram segments of medium size in both time and frequency". Lines 218-219 seem to suggest that, but the wording there is "intermediate-sized features", and I am not sure if this is supposed to be the same as "mid-level" features or something completely different.

3. I see a number of problems with the authors' claim of the importance of intermediate-sized features (l. 218), and especially with "intermediate length" (116-125).

In Fig. 6 the authors show that constraining to small-sized features (<1 oct, <100 ms) is detrimental to classification. But they do not test the effect of removing these small-sized features. Maybe that would affect classification as well, if both small and medium/large features are necessary.

Next, the word "intermediate" suggests that these features are considered more important in

comparison not only to small-sized features but also to large-sized features (say >4 octaves, >250 ms, or whatever thresholds would be appropriate). The effect of constraining to or removing large-sized features from the classification is not tested though. Finally, it would be interesting to vary feature size in *f* and *t* independently.

In l. 116-125 I have a problem with the claim that, based on merit distributions, integration over 100-200 ms is essential. The distributions (in suppl. Fig 1) use maximal merit, which would equate merit derived from a temporal window that consistently produces high merit with one that generally produces low merit except for one or two outliers. For example, based on the dot color in suppl. Fig. 1, it seems that for the twitter call the ideal window is more like 50-100 ms, even if max merit is similar in longer windows. I'd suggest adding a plot of average merit to the figure. And a color scale is absolutely necessary.

Also, Fig. 4 B suggests that for the trill the most informative features were shorter than 100 ms, while for the twitter and phee many were longer than 200 ms, up to 1 s in phee.

4. Matching MIF "neuron" properties to properties of actual A1 neurons (Fig. 7). Apart from panel A, the claim of matching properties is based on showing SINGLE example neurons with some properties similar to top MIFs. I do not find this convincing enough. These are top MIFs, so presumably there are many neurons showing similar properties. If that's true, this information should be provided!

Also, why are only twitter MIFs used for comparison with actual neurons? This raises the question if the finding generalizes across call types.

5. A few things are not clear regarding the number of MIFs per call type.

In l. 11 the authors say they needed "only ~10 features" for classification, which I read as "about 10". But Fig. 5 and the accompanying text say that exactly 10 were used. L. 142-143 mention 11, 20 and 18 features. So if 11-20 features were determined by the greedy search to be necessary, why were only 10 used subsequently?

More importantly, it is unclear how the authors decided on the number of MIFs per call type; in other words, what was the condition to finish the greedy search? L. 627 says "no further improvement in classification performance", but that's very vague. This is important as the necessary number of MIFs seems to be one of the main points the paper makes, and also arguments are made based on the number of MIFs in different circumstances (in classification of caller identification).

This brings me to the numbers in Fig 4A. If I read it correctly, for the twitter classification the first MIF provided almost perfect classification, and subsequent ones added 0.01 bits or less, so why add them, and then why stop at 11? Also for the phee, MIFs beyond the first one seem to add very little information. Only for the trill, MIFs beyond the first one seem to add something substantial. This seems to be indirectly confirmed by Fig 6B, where only for the trill the classification seems to be strongly impaired by choosing MIFs based on merit rather than added information. I interpret it that for the twitter and phee the MIFs beyond MIF #1 are of so tiny importance, that using non-optimal algorithm that is likely to select redundant features does not harm classification. (Interestingly, also only the trill classification seemed to be strongly affected by using waveforms instead of cochleograms; is this related in any way or just a coincidence?).

My bottom line is that the authors should thoroughly clarify the issue of the number of MIFs.

Minor issues

Greedy search algorithms, although computationally lightweight, have the disadvantage of possibly failing to find optimal solutions by getting stuck in local maxima. Given the performance of resulting MIFs and controls shown in suppl. Fig. 2, this is probably not an issue here, but it should be mentioned and briefly discussed. If the computational cost is not too high, providing at least one comparison to results obtained with an algorithm which does not have this problem, would be beneficial.

The argument for doing 1 vs all classifications instead of three-way is not terribly convincing (l. 605).

Whenever cross-correlations between MIFs and cochleograms are used, are these 1-D cross-correlations with leads/lags along time axis, or 2-D cross-correlations with leads/lags along both time and frequency axes? The latter does not seem likely for many reasons, but please make clear.

I am not sure if I understand the importance of MIF weights, they do not differ that much (Fig 4) but

are dutifully applied in various simulations.

l. 45 & Fig. 1. B are these 3 exemplars from different individuals, as suggested in l. 33-35? Then this is not showing within-individual variability, please adjust figure legend or otherwise clarify.

l. 71 "increasing resources" I guess you mean "increasing along the cortical processing hierarchy", or "along the superior temporal lobe"? Please clarify. Same in l. 104.

l, 71-74 I don't see why behavioral salience of calls and call selectivity in the temporal pole suggest that call classification is "a critical first step". One could even argue that – as selectivity does not really arise before the temporal pole – it's NOT the first step.

Also in line 105, based on the same premises, you say that "call processing is a computational GOAL of auditory cortical processing", so there seems to be a disconnect here.

l. 110 please say that for the purpose of the paper a "feature" is a randomly selected rectangular segment of a cochleogram limited in time and frequency (and not, for example, a parameter similar to those discussed around l. 480)

I would also like to see a justification why the features were rectangular and not, for example, elliptical (or weighed elliptical, e.g. 2D Gaussian).

l. 112 I think that the way in which you describe 'merit' here may be confusing, the one in l. 610 is clearer.

Fig.3 Are MIFs in this figure the first five MIFs from Fig. 4A? Please clarify.

l. 142-143 "xxx vs. rest task" vs. "xxx vs. all other calls" in l. 129-130 – I presume these are the same, please harmonize. Also "task" does not feel right in the context.

l. 147 I do not see how these MIFs capture harmonic relationships

l. 173 "neurons" please state that these are simulated neurons. Also line 183, state simulated responses.

l. 200 Maybe: Control simulations?

l. 261 and elsewhere: I believe it's guinea pig, not Guinea pig.

l. 276 one OR more?

l. 278 as with comparison to real neuron properties, why this is done using only twitters?

l. 285/6, 663 isn't "cross-correlation at lag 0" simply correlation?

l. 303 please do not abbreviate to "max. norm."

l. 309 "recordings"?

l. 388 what is meant by "four binary classification tasks", did you run classification between two callers in four pairs? If yes, why not 1 vs. all like with call type?

l. 391 Please explain why a small number of MIFs should indicate that calls differed in frequency

l. 482 Your study also uses different sets of parameters ("features") for different calls

l. 504 "non-mouse species" is rather awkward

Reviewer #3:

Remarks to the Author:

This study addressed general spectrotemporal features of marmoset vocalizations that are informative for tasks of classification (e.g. call type, caller identity), and found some intermediate sized parts of full characterizations to be most information bearing. That alone would be just acoustic feature analyses. Authors go further to address possible relationships of those features with neuronal responses in marmoset A1, by considering those features as spectrotemporal filters of artificial neurons. They show that responses of such simulated neurons exhibit acoustic tuning similar to those observed in actual single units of marmoset A1. Combining those results, authors conclude that intermediate parts of full spectrotemporal pattern of vocal sounds are most informative and what cortical neurons are tuned to.

My opinion is that the value of this manuscript is rather conceptual to find/use the informative acoustic features. I am somehow skeptical about whether it really is the neuronal mechanisms used in every species. On the other hand, their analyses may work well and bring similar conclusions for other

well studied vocalizations like bat emission for echo-location or human phonemes.

Even though the approach to address vocal call coding is novel, and appealing well to the topic of vocal coding, there are several concerns that should be clarified.

(1) Are those spectrograms starting at the onset of vocal sound? It is not specifically described. But I believe that it is very important to mention for this study.

Assuming those started at the onset of sound, were durations of sounds similar to one another, between and within call types or callers? If durations differ systematically, e.g. 800 ms vs. 1 s, then a feature of 200 ms starting after 800 ms can be very informative.

(2) For any classification tasks in the manuscript, I am not sure of the significance of any features (MIF) that start late. I guess that animals can find the call type (e.g. twitter or not) or caller identity (e.g. baby or adult) by the time such late features start. What are the onset timing of all primary features that are most informative?

(3) Any bias in vocal call set, like the number of calls per animal subject, separately for each call type? It is desirable if the numbers of calls are nearly even across animals.

Certainly, more subjects are better to obtain more general MIF. Thus, it is understandable if there is a bias of the number of calls between animals because the number of animals is limited. But it is better to clarify the compositions of data samples.

The most informative MIF of twitter looks like one particular call, while other twitters apparently look different, making me worry that the number of calls of twitter is dominated by one animal that was making twitter call look like the MIF.

Authors could test how exclusion of one animal's calls from dataset alters the MIF for call type classification task, how general the MIF is across animals.

(4) Similar to my comment just above, while authors examined the performance of MIF by cross validation using different dataset between learning and testing, it would be nice if authors did so also by using features of a group of callers to classify call types of different callers.

(5) When MIF differs in duration between datasets of call types, as seen in Fig 4, that may somehow relate to the temporal envelop modulation of sound. Isn't it worth commented in text?

(6) Figure 8. The title of the figure is "general applicability ... for other auditory tasks." But I am not sure if it is fair enough to say "general." "other auditory" are limited to caller identity or another species. Just "applicability ... for other tasks" sounds more consistent with the manuscript sense to me.

Do authors already have actual neuronal responses that can classify calls?

(7) Caller identity difference may relate more to F0. I may guess narrower bandwidths of MIF. Authors could do plots like Figure 4B for the MIF derived for the task in Figure 8A, if it reveals task-dependent differences of MIF.

(8) If I adopt authors' arguments about MIF representation in A1, I speculate that there are A1 neurons equipped to code MIF. Since cochleographic representations of MIF are somehow equivalent to STRF, it would be nice to see some STRF measured for A1 neurons have similarity to MIF or partial overlaps with MIF. Similarly, I guess there are neurons tuned to lower frequencies in the region where MIF has vacant power as in the figure 3C, 2nd column. It would be nice to know whether such actual neurons, not MIF, remained silent during call sounds.

(9) When simulated firing by MIF (Fig. 5), what timing were the MIF spike time relative to the period of MIF? Timing could be onset, center or offset of MIF. The durations differ between MIF. So, do long MIF need to wait longer to fire a spike? Are there A1 neurons that respond to calls in such way, too? Some MIF seem well exclusive to particular call type. Do actual neurons behave that way?

This consideration may be more difficult when it comes to caller identity. Do some actual neurons respond exclusively to particular caller?

(10) Random features extraction. This question is somehow obsolete, not affecting conclusions. But, it was not clear how.

It seems every feature was a rectangle area of cochleograms, it would be defined by 4 variables (2 for position and 2 for size). I can imagine that such 4 variables can be swept systematically, from single pixels to the whole, to find most informative rectangle. Does random generation of rectangles mean random selections of those variables independent of one another?

(11) Figure 7C. It is interesting to see peaks of firing rate response at 60 and 100 ms and a trough at 80 ms for neurons (top), and MIF (bottom) share such peaks and appear scarce at the trough. Any reason why?

(12) Figure 4A. I think mutual information (merit) and log-likelihood ratio (weight) usually are monotonically related. Correct?

Am I correct about my description below?

"Added information are small for MIF after the first one. It is because the first column shows additional information after MIF in rows above added information. However, mutual information of MIF, that could be between 0 and 1, are high compared to added information, because many MIF have redundant information. Larger LL-ratio means the mutual information being informative is significant."

I think text could describe such details of how to interpret those parameters.

When MIF position/size differed, how redundant are they about mutual information?

(13) Several words deserve more explicit descriptions.

What is "normalized cross correlation"? Sometimes it is not clear cross correlation between what. It is entirely not clear what was normalized by what. Only description of it in the text is "commonly used metric to quantify template match."

What is "power law nonlinearity"? It could be anything with $y = Ax^n$ with any n other than $n=1$. Texts in Results direct readers to see Methods. But, in Methods, it is said only "that was scaled arbitrarily". What was scaled arbitrarily is not clear either. The power? The proportionality?

(14) Figure 7 legend. "B-top" should be "A-top," and "C-E" should be "B-D."

(15) "one of more of MIF"

Response to reviewers

We thank the reviewers for their time and effort in providing extensive feedback on our manuscript. We have now addressed all reviewer questions which has led to substantial additions to the manuscript. In addition to clarifications and analysis of previously presented data, we now present new data, including: 1) a detailed characterization of production variability in marmoset calls that helps clarify some reviewer questions, 2) new applications of the model for categorizing guinea pig calls and macaque calls, 3) new analysis confirming the usefulness of “intermediate” features, and 4) new single-unit data from marmosets and guinea pigs demonstrating examples of feature selectivity in-vivo. The manuscript has been significantly improved as a result of these suggestions, and we are grateful to the reviewers for their contribution.

We have added Pilar Montes-Lourido, who contributed in-vivo recordings from guinea pig auditory cortex and played a central role in recording and categorizing guinea pig calls, as a second author to the manuscript. We obtained additional data (macaque vocalizations) from Dr. Yale E. Cohen, who is now acknowledged in the manuscript with his permission. We also acknowledge students who contributed to the collection of guinea pig vocalizations in the revision.

In this document, the reviewer queries are in green font, our responses are in black font, and additions to the revised manuscript are in blue font. We also present some additional data in response to reviewer queries in this document, which we do not include in the manuscript for the reasons stated.

Addition not requested by reviewers:

A recent study that used statistical classifiers on call parameters (such as fundamental frequency, duration etc.) and the modulation spectrum to achieve caller identification for macaque coo calls came to our attention (Fukushima et al., Ref. 45). As this study was highly relevant to ours, we have included a discussion not requested by the reviewers.

Page 32 / Line 661: “In a recent study, a combination of the above approaches was used in conjunction with statistical classifier techniques to achieve caller identification for macaque coo calls⁴⁵. Caller identification could not be achieved using a single feature alone, where feature referred to a parameter such as fundamental frequency, duration, or location in the modulation spectrum. Rather, a combination of cues was required for high caller identification performance. Our study differs from this study in that our definition of ‘feature’ is non-parametric, our goal is to generalize over individual identity, and features are contrastive and task-dependent. But similar to this study, a single feature alone was insufficient for call categorization in our study as well.”

Reviewer #1 (Remarks to the Author):

This paper clearly demonstrates that production-invariant classification of calls can be achieved using mid-level acoustic features. This is an important result and would be of interest to a wide audience. The paper was a joy to read. The results and the analyses were presented convincingly. I applaud the authors for the clarity of their writing.

- We thank the reviewer for these very encouraging remarks!

My major objection is to the title of the paper. Auditory recognition is much broader than call classification, which is the main focus of the paper, and caller identification. I suggest to change the title to one that reflects the scope of the paper.

- We agree, and have changed the title of the paper to “Optimal features for auditory categorization”.

Other minor comments are:

1) The auditory nerve model that transforms calls into cochleagrams use human auditory nerve parameters. Please comment on the applicability of these model to marmosets.

- The auditory nerve model that we used is available for cats and humans. This depends on the availability of actual auditory nerve recordings, or the availability of fine-grained psychoacoustics data from which nerve fiber bandwidths can be estimated. Such data is not currently available for marmosets. We chose to use the human version because the marmoset audiogram closely resembles the human audiogram (ref. 54). This is now justified in the text at **Page 37 / Line 772**: “We used human auditory nerve parameters because of the close similarity between marmoset and human audiograms⁵⁵”

2) Please provide more details to how you estimate mutual information.

- We did not provide this detail earlier because we used the method of Ullman et al., 2002 without any adaptation. In the revision, we now provide additional details:
- **Page 38 / Line 805**: “We computed mutual information following the method of Ullman et al.¹², by measuring the frequency of detecting a feature f_i at a given threshold θ_i ($f_i = 1$ if present, 0 if absent) in the within-class ($C=1$) or outside-class ($C=0$) cochleagrams as:

$$I(f_i(\theta_i), C) = \sum_{\substack{f_i = \{0, 1\} \\ C = \{0, 1\}}} p(f_i, C) \log \left(\frac{p(f_i, C)}{p(f_i)p(C)} \right)$$

where $P(C)$ was assumed to be 0.10. We empirically verified that features identified were insensitive to variations of this value.”

3) Color bar is missing in Supp. Fig. 1

- A color bar has been added to Supp. Fig. 1 (now Supp. Fig. 2).

4) Figure 1C is hard to interpret.

- We have removed this figure from the revision, as the information that is needed to improve the interpretation of the figure is previously published (Fig. 7 of Sadagopan et al., 2015). In that publication, the interpretation of this figure is aided by other panels.

5) Could the authors comment on the relationship of their work to "Unsupervised learning of invariant representations" (doi: 10.1016/j.tcs.2015.06.048). This work also uses random features extracted from stimuli to build invariant representations, however in an unsupervised way.

- We thank the reviewer for bringing this truly interesting article to our attention. The level of theoretical sophistication necessary to truly appreciate this model is quite high, and we

apologize in advance for any erroneous generalizations made here concerning our discussion of this model.

- From our understanding, this model derives image 'signatures' by applying a set of nonlinear transformations to randomly chosen templates. The signatures serve as the representation of the probability distribution of the result of transformations acting upon an image. These signatures show invariance over a given set of image transformations, and can achieve a high degree of selectivity. These computations can be implemented using simple architectural motifs, and by layering these motifs, high levels of performance can be achieved in naturalistic data sets. This architecture resembles the organization of visual cortex. This framework predicts that a similar mechanism could apply to auditory cortex.
- It is tempting to relate the MIFs of our model to the 'signatures' of this model. In both cases, some core feature that can generalize over variability is learned. From statistical analysis of marmoset vocalizations, it is known that the set of vocalizations can indeed be thought of as a multivariate probability distribution (Agamaite et al., 2015). From our model, we show that an MIF generalizes over this distribution, and can be thought of as a proxy for this distribution. At face value, both the MIF and signature pick up the 'gist' of an image, and changes around this gist are tolerated. Both the MIFs and signatures can be learned from small sample sizes. We show preliminary evidence that the MIFs might arise in A1 superficial layers, somewhat similar to 'complex' cells that might be implementing the 'signatures'.
- However, key differences arise in learning. The MIFs are by definition contrastive – they are designed to pick up critical differences between sounds. In the broadest sense, if conceptualized as the difference between sound A and all other sounds, perhaps the MIF is truly picking up an inherent image or sound feature. However, learning the MIFs requires supervision, i.e., access to the class identity of the training exemplars.
- An extensive discussion of this model is beyond the scope of our manuscript, but we have briefly discussed it as follows. We are happy to edit our description if the reviewer has any feedback.
- **Page 31 / Line 625:** “Recently, theoretical efforts have been directed at learning invariant representations from small training sets using unsupervised methods⁴⁰. In this model, image 'signatures' which serve as a proxy for the probability distribution of an image and its transformations are learnt by leveraging the time correlations of image transformations in the real world to label image identity. Image signatures can be computed by complex cell-like units using Hebbian learning rules. This model predicts that a similar computation might occur in auditory cortex. The MIFs that we have derived for call categorization are similar to the image 'signatures' in that they serve as a proxy for the probability distribution of a sound category that has been subjected to production variability. Indeed, vocalizations can be viewed as multivariate probability distributions along multiple call parameters⁴¹, and MIFs could serve as the 'gist' of a call category around which these variations occur. Similar to image signatures, MIFs seem to be computed by superficial-layer auditory cortex neurons. However, differences arise in how MIFs are learnt. Although small sample sizes are adequate, unlike image 'signatures' that are learnt by observing image transformations over time, explicit labeling of the class of input examples is necessary for learning the MIFs of calls. Conceptually, whereas image 'signatures' are learnt by observing within-category transformations, MIFs are learnt by contrasting the distributions of sound categories.”

Reviewer #2 (Remarks to the Author):

Major concerns

1. The methodology of extracting meaningful “features” and the resulting high classification accuracy look impressive at first glance. But on closer inspection it appears to boil down to “if we select those parts of a call representation that distinguish very well between the calls, and then use them to distinguish between the calls, they distinguish very well”. Maybe I am missing a crucial point here, but then the authors should probably make this point clearer.

- The overall question of this study is “How can you classify calls, vocalized differently by different animals, into discrete call categories?” Although there is a rich literature of ‘vocalization selective’ responses at various stages of the auditory pathway, there has been little theoretical investigation into *how* this selectivity may be achieved. Our study proposes a novel theoretical framework within which vocalization processing can be investigated.
- The specific hypothesis we are testing to answer our overall question is, “Can you classify calls made differently by different animals into call categories using smaller features that best distinguish call categories?” In this study, we show that we can indeed, by selecting parts of call representations that maximally distinguish calls. We go on to ask what types of features are appropriate (‘intermediate lengths’ or ‘mid-level’), how many we would need (10-20 MIFs), what algorithms may be used to derive these features (greedy search and information maximization), whether feature-based approaches generalize to other tasks (yes), and whether any experimental evidence supports the existence of such a strategy in the brain (yes).
- We think that the reviewer’s confusion is arising because the reviewer is perhaps taking a limited view of the ‘training’ and ‘testing’ steps of the model. The testing step of the model is to be viewed as a validation step that is common to many, if not all, computational models. It demonstrates that whatever optimization we have done is not ‘over-fit’ to the training stimuli, and that the model will work as intended when confronted with new inputs. But the reviewer is also correct in pointing out that this needs to be clarified in our writing. We have made the following changes in the revision to address this question.
- Page 5 / Line 80: “Although a number of studies have described call-selective responses at various stages of the auditory pathway, there has been little investigation into how the auditory system goes about solving these problems, both at the algorithmic and mechanistic levels. In this study, we started with the premise that the detection and classification of calls into discrete call types is a critical first step that enables the above computations. Our overall question in this study was to ask how production-invariant call classification could be accomplished in the auditory pathway. Specifically, we tested the hypothesis that production-invariant call classification could be accomplished by detecting constituent features that maximally distinguish between call types. Starting from an initial set of randomly selected marmoset call features, we used a greedy search algorithm to determine the most informative and least redundant set of features necessary for call classification. We show that high classification performance can indeed be achieved by detecting combinations of a small number of mid-level features.

2. The authors emphasize that the classification is based on “mid-level features” but they never state clearly what they consider “mid-level” features, and, by extension, what would be “low-level” or “high-level” features. It might be that the authors simply mean “cochleogram segments of medium size in both time and frequency”. Lines 218-219 seem to suggest that, but the wording there is “intermediate-sized features”, and I am not sure if this is supposed to be the same as “mid-level” features or something completely different.

- We thank the reviewer for bringing this up, and we agree that we were unclear on what we considered 'mid-level' features. What we would like to mean by 'mid-level' features is actually features of intermediate complexity, as is the case for face detection in vision. For example, a pure tone (a line on the cochleagram) or noise would be a 'low-level' feature, whereas an entire vocalization would be 'high-level'. In this conceptual scheme, 'mid-level' would refer to features such as FM sweeps, phrasic structure, etc. However, because quantifying feature complexity was tricky, we had operationally defined 'mid-level' to mean 'intermediate-sized'. We agree that this was not the ideal definition.
- In the revision, we now use the reduced kurtosis of the activity distribution of the auditory nerve fibers contained in a feature as a metric of feature complexity (**updated Suppl. Fig. 2**). As we explain in the text,
- **Page 8 / Line 154:** " We characterized feature complexity using the reduced kurtosis of the activity distribution of all auditory nerve fibers contained within a feature. Briefly, if the feature was an 'empty' region of the cochleagram, or a region of uniform activity, the activity of all nerve fibers in all time bins would be about equal. This activity would thus be normally distributed, and show a reduced kurtosis value of zero. At the other extreme, for entire calls, there would be many bins of high activity, and a large number of bins with zero activity, resulting in an activity distribution with very high reduced kurtosis. We hypothesized that 'mid-level' features that represent aspects of calls such as frequency-modulated sweeps or combinations of phrases over time would show intermediate reduced kurtosis values, and be more informative than 'low-level' (tones) or 'high-level' (entire calls) features. Consistent with this idea, we found that while features of low merit showed low kurtosis values and whole calls showed high kurtosis values, features of high merit showed intermediate kurtosis values, supporting the hypothesis that 'mid-level' features of intermediate complexity were most informative for classification (Suppl. Fig. 2)."

3. I see a number of problems with the authors' claim of the importance of intermediate-sized features (l. 218), and especially with "intermediate length" (116-125). In Fig. 6 the authors show that constraining to small-sized features (<1 oct, <100 ms) is detrimental to classification. But they do not test the effect of removing these small-sized features. Maybe that would affect classification as well, if both small and medium/large features are necessary.

- Please see answer to the next question below.

Next, the word "intermediate" suggests that these features are considered more important in comparison not only to small-sized features but also to large-sized features (say >4 octaves, >250 ms, or whatever thresholds would be appropriate). The effect of constraining to or removing large-sized features from the classification is not tested though. Finally, it would be interesting to vary feature size in f and t independently.

- We thank the reviewer for these specific suggestions on how to appropriately support our claim regarding the precedence of intermediate-sized features. We now present several additional analyses in **new Fig. 7** that addresses this point. In this figure, we systematically test the effects of using only small or removing small features (<1 oct., < 100ms), using only large and removing large features (>2 oct., > 250 ms), using only high or low bandwidth features (independent of time), using only short or long features (independent of bandwidth), and using features of different complexity values from the initial set of randomly generated features. We have also analyzed performance using the largest possible features (whole calls) and found it to be quite poor compared to the default model.

- These new analyses show that the default model indeed performs better compared to any of these alternate models. This is presented in the revision under a new section in the Results section.
- **Page 18 / Line 340:** “**The precedence of intermediate-sized features for classification**
We have previously shown that features of intermediate lengths and complexities possess high individual merits for classification (Supp. Fig. 2). We have also shown that the set of MIFs is composed mainly of features of intermediate lengths relative to the entire call (Fig. 4B). To directly test whether features of intermediate size were indeed the most informative, we re-derived MIFs after constraining the initial set of features to particular time and frequency bins and quantified model performance (Fig. 7). When we constrained the features to be only small (<100 ms and <1 oct.) or removed all small features, performance was worse than the default model (Fig. 7, top row). Similarly, model performance was worse when we constrained to large features (>250ms and >2 oct.), or removed all large features compared to the default model. When we constrained bandwidth and time independently to be large or small, model performance was worse compared to the default model, with large values being more detrimental (Fig. 7, bottom row). As previously discussed, using the largest possible features (whole calls or average call) resulted in poor classification performance as well. These results demonstrate that features of intermediate size indeed provide the best classification performance.”

In l. 116-125 I have a problem with the claim that, based on merit distributions, integration over 100-200 ms is essential. The distributions (in suppl. Fig 1) use maximal merit, which would equate merit derived from a temporal window that consistently produces high merit with one that generally produces low merit except for one or two outliers. For example, based on the dot color in suppl. Fig. 1, it seems that for the twitter call the ideal window is more like 50-100 ms, even if max merit is similar in longer windows. I'd suggest adding a plot of average merit to the figure. And a color scale is absolutely necessary.

- We used maximal rather than average merit in the marginal histograms in Supp. Fig 1 (now Supp. Fig 2) because regardless of the length of the feature, features could be completely uninformative if they do not overlap at all with the frequency bands of any of the calls. The same is true for bandwidth – regardless of any bandwidth, the feature may select a portion of the call during which the envelope is zero (such as the inter-phrase interval in twitters). Averaging over all features would therefore be misleading. By using the maximum in each bin, we were comparing the best possible feature in each time bin. We completely agree with the reviewer that this would “flatten” the curves, but we believe that this is a better comparison.
- Regardless of what happens at the level of the initial random features, Fig. 4B shows quite clearly that intermediate lengths, especially when viewed relative to the call lengths, are over-represented in the MIF set. In the revision, we have reworded our description of the initial feature set (6000 features in S.Fig. 2), and restricted the discussion of actual feature lengths to the MIFs in Fig. 4B. We provide MIF lengths and call lengths as well. The following edits to the text clarify these points:
- **Page 8 / Line 143:** “In Supplementary Fig. 2, we plot the merits of all 6000 initial features as a function of each feature's bandwidth and temporal integration window. Along the margins, we plot the maximum merit of features within each bandwidth- or temporal window bin. **These distributions compare the best features from each time bin, and show that**

features of intermediate lengths relative to the total call length show higher merits for call classification. This is an expected consequence of two characteristics ...”

- **Page 11 / Line 216:** “In frequency, MIFs neither encompassed an entire call, nor consisted of only few frequency bands. In time, MIFs showed integration windows of the order of hundreds of milliseconds (Fig. 4B). The mean MIF lengths were 215 ms, 68 ms, and 406 ms for twitters, trills, and phees respectively. Compared to the average lengths of the calls (twitters: 1.25 s, trills: 0.5 s, phees: 1.27 s), these correspond to 17%, 14%, and 32% of mean call length respectively. Interestingly, these lengths may correspond to time scales of temporal modulations in calls – for twitters, the sum of mean phrase length and mean inter-phrase interval is ~190 ms; for trills, the mean amplitude modulation period is ~30 ms. Thus, features of intermediate lengths were especially informative for call classification.”

Also, Fig. 4 B suggests that for the trill the most informative features were shorter than 100 ms, while for the twitter and phee many were longer than 200 ms, up to 1 s in phee.

- We agree with the reviewer, and realize that using 100-200ms as a shorthand description of the order of lengths created confusion. The mean MIF lengths were 215 ms, 68 ms, and 420 ms respectively for twitters, trills and phees. The corresponding mean call lengths are 1.25 s, 0.5 s, and 1.27 s. Therefore, in the revision, we have reworded to:
- **Page 11 / Line 216:** “In frequency, MIFs neither encompassed an entire call, nor consisted of only few frequency bands. In time, MIFs showed integration windows of the order of hundreds of milliseconds (Fig. 4B). The mean MIF lengths were 215 ms, 68 ms, and 406 ms for twitters, trills, and phees respectively. Compared to the average lengths of the calls (twitters: 1.25 s, trills: 0.5 s, phees: 1.27 s), these correspond to 17%, 14%, and 32% of mean call length respectively. Interestingly, these lengths may correspond to time scales of temporal modulations in calls – for twitters, the sum of mean phrase length and mean inter-phrase interval is ~190 ms; for trills, the mean amplitude modulation period is ~30 ms. Thus, features of intermediate lengths were especially informative for call classification.”

4. Matching MIF “neuron” properties to properties of actual A1 neurons (Fig. 7). Apart from panel A, the claim of matching properties is based on showing SINGLE example neurons with some properties similar to top MIFs. I do not find this convincing enough. These are top MIFs, so presumably there are many neurons showing similar properties. If that’s true, this information should be provided! Also, why are only twitter MIFs used for comparison with actual neurons? This raises the question if the finding generalizes across call types.

- We understand the reviewer’s concern. The model described here was developed quite recently, whereas the data that we compared to were gathered several years ago. Since experiments are designed with specific questions in mind, we were restricted to whatever appropriate data we could re-use appropriately for our comparisons. In panel A for example, the original experiments were designed to probe species specificity of vocalization processing by looking at forward and reverse call representations in marmoset and cat auditory cortex. Because of its phrasic and directional spectrotemporal structure, twitter calls were used for this purpose. This is why our comparison was restricted to twitters.
- Another study from which we derived data was a study of combination sensitivity (Sadagopan and Wang, 2009). In that study, we had gathered responses to linear FM and bandpass noise stimuli because those were stimuli needed to validate predictions of a

combination sensitivity model. Since twitters are made of linear FM-like elements, once again, the examples that we could find were also related to the twitter call.

- These comparisons of feature 'responses' and cortical responses represent a first step towards verifying the model presented here. In the revision, we 1) made some edits to indicate that these were first steps, and 2) present additional data in new Fig. 9.
- **Page 2 / Line 11:** "Most importantly, predictions of the tuning properties of putative neurons selective for such features accurately matched some previously observed responses of superficial layer neurons in primary auditory cortex."
- **Page 21 / Line 415:** "MIF tuning properties match some single unit recordings from A1 L2/3
So far, we have demonstrated that classification based on MIFs derived purely using theoretical principles can achieve high levels of production-invariant call categorization. We then asked if the auditory system uses such an optimal feature-based approach to call classification. To explore this possibility, as a first step, we generated 'tuning curves' of putative MIF-selective model neurons responding to commonly used acoustic stimuli and asked if these tuning curves matched previous experimental observations. In this effort, we were restricted by the appropriateness and availability of previous data. To do so...'
- **Page 23 / Line 463:** "Most strikingly, we could recapitulate some specific and highly nonlinear single-neuron tuning properties as well..."
- To rigorously test the model, experimental design would directly get at what encoded features are and how they compare to theoretically predicted features, while controlling for low-level effects. These are precisely what we are currently working on using guinea pigs as a model system. These data have been presented at conferences (SfN 2017, Gordon Conference 2018, to be presented at ARO 2019), and we estimate that these experiments are about 70% complete, but we would require significant additional time to complete the analysis. Therefore, while we have decided to include a few more single neuron examples supporting the model in this revision, as we will explain below, for us to properly describe the experiments and necessary controls, we would need at least 4 additional figures and 2-3 more pages of text, which would complicate an already rich manuscript (10 main figures and 7 supplementary figures in this revision).
- Recall that at the MIF level, false alarms and missed detections are possible, which is why individual MIFs do not have 1 bit of information. True call categorization occurs by taking a weighted sum of MIF responses. In recordings from both marmoset and guinea pig A1, we do find neurons that appear to be selective for features across 1 or 2 call categories. Three examples from each species are shown in new Fig. 9.
- **Page 24 / Line 483:** "Consistent with the prediction of feature selectivity, we have found neurons in A1 of both marmosets and guinea pigs that respond selectively to conspecific call features. In Fig. 9, we present the spike rasters of example single neurons in both marmoset and guinea pig A1 responding to marmoset (Fig. 9A) and guinea pig calls (Fig. 9B) respectively. We presented multiple exemplars of each call type as stimuli. These example neurons responded at specific time points to a few call stimuli, typically across 1 – 3 categories. Such responses are consistent with our feature-based model because single features alone do not completely categorize calls, i.e., MIFs do not have 1 bit of information for categorization. Rather, combinations of features weighted by their log-likelihood ratios are necessary to ultimately achieve complete call category information. These data provide promising support for our model, but further experiments are necessary to: 1) determine

how informative these neural features are about call category and how they compare with model features, 2) to confirm where such responses arise in the auditory pathway, and 3) to account for possible low-level confounds. Experiments are presently ongoing to address these issues.”

- Here, we also briefly summarize our ongoing experiments, not to be included in the manuscript, so that the reviewers may appreciate: 1) the efforts that are underway to directly test the model, and 2) why presenting all these data would require much more space and may exceed the scope of this manuscript.

- [Redacted text block]

- | [Redacted text block]

- | [Redacted text block]

- | [Redacted text block]

[REDACTED]

[REDACTED]

[REDACTED]

[REDACTED]

[REDACTED]

5. A few things are not clear regarding the number of MIFs per call type. In l. 11 the authors say they needed “only ~10 features” for classification, which I read as “about 10”. But Fig. 5 and the accompanying text say that exactly 10 were used. L. 142-143 mention 11, 20 and 18 features. So if 11-20 features were determined by the greedy search to be necessary, why were only 10 used subsequently?

- We apologize for the confusion and inconsistent terminology which we have clarified in the manuscript as well as in response to the questions below. We used “~10” in the abstract to describe the order of magnitude of the number of features required across the tasks we had tested. The numbers of fragments were 11 (twitter), 20 (trill) and 16 (phee, was erroneously stated as 18). In Fig. 5, we showed the response of the top 10 of (11, 20, or 16) features purely for uncluttered visualization purposes. When the performance of the model is evaluated using ROC analysis (Fig. 6), we use all MIFs. This is now made clearer in the text.
- **Page 2 / Line 10:** “Call classification at >95% accuracy could be accomplished using only 10 – 20 features per call type”
- **Page 13 / Line 241:** “In Fig. 5, we plot the spike rasters of simulated MIF-selective neurons for twitter, phee, and trill (top 10 MIFs shown), responding to a train of randomly selected calls from the novel test set.”
- **Page 13 / Line 251:** “We quantified the performance of the entire set of MIFs (n=11, 16, and 20 for twitter, phee, and trill respectively) for the classification of novel calls by ...”
- **Fig. 4B** updated to include all MIFs.

More importantly, it is unclear how the authors decided on the number of MIFs per call type; in other words, what was the condition to finish the greedy search? L. 627 says “no further improvement in classification performance”, but that’s very vague. This is important as the necessary number of MIFs seems to be one of the main points the paper makes, and also arguments are made based on the number of MIFs in different circumstances (in classification of caller identification).

- We added features until we could no longer increase hit rate without increasing false alarm rate. For practical reasons, we stopped adding features when we achieved a total information content of 0.999 bits, or when individual features added less than 0.001 bits of information, whichever was reached first. This is now stated in the methods.
- **Page 40/ Line 834:** “We iteratively added MIFs until we could no longer increase the hit rate without increasing the false alarm rate. Practically, this meant adding features until total information reached 0.999 bits, or individual features added less than 0.001 bits, whichever was reached earlier.”

This brings me to the numbers in Fig 4A. If I read it correctly, for the twitter classification the first MIF provided almost perfect classification, and subsequent ones added 0.01 bits or less, so why add them, and then why stop at 11? Also for the phee, MIFs beyond the first one seem to add very little information. Only for the trill, MIFs beyond the first one seem to add something substantial. This seems to be indirectly confirmed by Fig 6B, where only for the trill the classification seems to be strongly impaired by choosing MIFs based on merit rather than added information. I interpret it that for the twitter and phee the MIFs beyond MIF #1 are of so tiny importance, that using non-optimal algorithm that is likely to select redundant features does not harm classification.

- As explained above, we added features until the total information content reached 0.999 bits, or individual features added less than 0.001 bits of information. For twitters, the authors interpretation is correct – we reach 0.95 bits of information, and therefore essentially solve the task, using the top feature. But for phees, note that the information added by the first feature alone is only 0.78 bits. Although successive features are informative by themselves (second column), they add a small number of new ‘hits’ without introducing additional false alarms. By the time all features are added, the total information for classification is around 0.9 bits. The same is true for trills.
- Over all tasks, the amount of information added by the first MIF and subsequent MIFs are: Marmoset calls: Twitter – 0.95 + 0.04 added, Phee – 0.78 + 0.12 bits added, Trill – 0.60 + 0.39 added. Guinea pig calls: Wheek (12 MIFs) – 0.9 + 0.09 added, Whine (9 MIFs) – 0.68 + 0.30 added, Rumble (3 MIFs) – 0.9 + 0.1 added. For macaque calls: Coo (5 MIFs) – 0.38 + 0.35 added, Grunt (4 MIFs) – 0.66 + 0.22 added, Harmonic Arch (9 MIFs) – 0.36 + 0.61 added. For this reason, we respectfully disagree with the reviewer’s interpretation - subsequent features do substantially contribute to the solution of the tasks.
- In the revised text, we now explain in **Page 11 / Line 202**: “Note that 1 bit of information corresponds to perfect classification. For twitters, detecting a single feature (the top MIF) was sufficient to gain 0.95 bits of information. Subsequent features probably detected only a few additional twitters without introducing new false alarms. For the other call types, however, the top MIF only provided 0.78 or 0.6 bits of information. Although successive MIFs individually had high merit (second column), they added little information to the top MIF (first column), likely because of redundancy – each MIF could only add a small number of additional ‘hits’ without introducing new false alarms. However, detecting these features was crucial for solving the task, as they ultimately elevated the total information to > 0.9 bits.”

(Interestingly, also only the trill classification seemed to be strongly affected by using waveforms instead of cochleograms; is this related in any way or just a coincidence?).

- We believe that this is coincidental. The strongest cue when using waveforms is envelope shape. Twitters have distinctive envelopes because they are a multi-phrase call, and this likely explains the high performance of both the cochleogram and waveform versions of the model. Phees and Trills have subtler envelope modulations, and therefore do not have this advantage when using the waveform. In this case, fine structure dominates, and as we mention in the manuscript, seems to be detrimental to performance.

My bottom line is that the authors should thoroughly clarify the issue of the number of MIFs.

- In addition to the above clarifications, we have provided the number of MIFs for all simulations performed in the paper. These are mentioned in:
- **Page 26 / Line 527**: “We applied the greedy-search algorithm to determine the MIFs for caller identification in a caller A vs. all other callers task (Fig. 10A). We found that similar to call categorization, caller identification could also be achieved using a small number of MIFs ($n = 4$).”
- **Page 27 / Line 542**: “In Supplementary Fig. 7, we plot the ROC for caller identification between a pair of marmosets with overlapping dominant frequencies. The MIF-based approach ($n = 20$ MIFs) achieved >80% hit rates with <10% false alarm rate for caller identification.”

- **Page 27 / Line 549:** “We used the MIF-based approach to classify guinea pig call types (‘whine’, ‘wheek’, and ‘rumble’) from all other guinea pig call types. Similar to marmosets, guinea pig classification could be accomplished using a handful of features (12, 9, and 3 MIFs for whine, wheek, and rumble), and MIF-based classification achieved high performance levels (Fig. 10B). Similarly, we implemented the MIF-based algorithm to classify macaque calls (using 5, 4, and 9 MIFs for coos, grunts and harmonic arches) from a limited macaque call data set³³ and achieved high classification performance (Fig. 10C).”

Minor issues

Greedy search algorithms, although computationally lightweight, have the disadvantage of possibly failing to find optimal solutions by getting stuck in local maxima. Given the performance of resulting MIFs and controls shown in suppl. Fig. 2, this is probably not an issue here, but it should be mentioned and briefly discussed. If the computational cost is not too high, providing at least one comparison to results obtained with an algorithm which does not have this problem, would be beneficial.

- We agree that greedy search has the disadvantage mentioned by the reviewer. We did not run other algorithms for two reasons: 1) as the reviewer mentioned, we checked and were confident that we were not getting stuck in local maxima as we could arrive at a similar solution from multiple starting points, and 2) we were not aware of other automatic call classification algorithms for marmosets at the time. However, we have since found one additional study that used a variety of machine learning methods for classifying marmoset calls. The study is based on a limited number of calls (Ref. 19), and attains a maximum accuracy of about 85%, but the model is not intended to be a biologically plausible model of call classification. We now compare the greedy search model to these alternate methods in the revised manuscript.
- **Page 19 / Line 373:** “In this study, we used greedy search and pairwise maximization of information to find optimal features. However, it is possible that the greedy search algorithm does not find an optimal solution because of its inability to overcome local maxima. We do not think this is the case because: 1) the model performs at high accuracy levels, leaving little room for significant improvements, 2) we could arrive at similar sets of MIFs and achieve similar performance levels from different initial feature sets, specifically when highly informative features were excluded (Supp. Fig. 3), and 3) we could match or outperform other machine learning based algorithms for marmoset call classification¹⁹. Therefore, the implemented greedy search algorithm likely converges at a true optimal solution.”

The argument for doing 1 vs all classifications instead of three-way is not terribly convincing (l. 605).

- We apologize for the very brief argument in favor of doing multiple 1 vs. all classifications. The pros and cons of doing (1 vs. all) or n-way classifications was in fact the central question addressed by the paper that we cited (Ref. 55, Akselrod-Ballin and Ullman, 2008). That paper showed that setting up the problem as multiple binary classification tasks produced distinctive features that are highly tolerant to distortions and missing inputs, whereas multi-way classifications results in larger features that were not as robust. They concluded that “The results show the advantage of distinctive features for making fine distinctions in a robust manner”, where ‘distinctive features’ is how they characterize features derived using multiple binary classifications. Based on that study, we had decided to set up our task as multiple binary classifications as well. In the revision, we now provide a more detailed explanation at a more appropriate location in the text.

- **Page 36 / Line 759:** “That is, three main **binary** classification tasks – *twitter* vs. all other calls, *trill* vs. all other calls, and *phee* vs. all other calls were considered. We set up the categorization tasks as a series of binary classifications (Twitter vs. all other calls, Trill vs. all other calls, etc.) based on the results of an earlier study of visual categorization that demonstrated the advantages of features learnt using multiple binary classifications compared to those learnt using a single multi-way classification. Specifically, in that study, multiple binary classifications resulted in features that were distinctive and highly tolerant to distortions⁵⁶.”

Whenever cross-correlations between MIFs and cochleograms are used, are these 1-D cross-correlations with leads/lags along time axis, or 2-D cross-correlations with leads/lags along both time and frequency axes? The latter does not seem likely for many reasons, but please make clear.

- Yes, the reviewer is correct in that these are 1-D cross-correlations with leads/lags along the time axis only. In other words, the feature can only be found in its frequency range, but can be found anywhere in time in a call.
- This is clarified in **Page 38 / Line 790:** “We effectively implemented a one-dimensional version of NCC by only considering the auditory nerve fibers that overlapped between the call and the feature. Note that this means features can only be detected in the frequency range that they span, but can be detected anywhere in time within a call.”

I am not sure if I understand the importance of MIF weights, they do not differ that much (Fig 4) but are dutifully applied in various simulations.

- We realize that there has been a lack of clarity on the distinction between ‘merit’ and ‘weight’. Merit and weight are subtly different quantities that need not be monotonically related. For example, it is possible that the **lack** of power in a certain frequency band is highly informative of a particular call type. A feature in this frequency band would have high merit for classification because it will reliably not be detected in the target category. But the feature would have a negative log-likelihood ratio as it is found more often in ‘other’ calls and not in the target category. We now state this in the Methods section.
- **Page 39 / Line 817:** “Note that merit and weight are distinct quantities that need not be monotonically related. For example, if the *lack* of energy in a frequency band is indicative of a target category, features that contain energy in this frequency band will be detected often in the other categories, but not in the target category. The feature will thus have high merit for classification, as it is informative by its absence, but have a negative weight.”
- We also use merit and weight for different purposes – merit is used for the selection of MIFs, whereas weight is used for simulating MIF responses. This is because once a feature is chosen, we are only interested in the amount of evidence a spike provides as to the presence of a category, and weight is a more direct metric of this.

I. 45 & Fig. 1. B are these 3 exemplars from different individuals, as suggested in I. 33-35? Then this is not showing within-individual variability, please adjust figure legend or otherwise clarify.

- The exemplars are indeed from three different individuals. The figure legend has been adjusted to clarify this. We have added new analyses in **new Fig. 1C-F and new Supp. Fig. 1** that demonstrates both within- and between-individual variability.

- **Page 7 / Line 129:** “Marmoset twitters can be characterized along several acoustic parameters such as bandwidth, duration, dominant frequency, and inter-phrase interval⁸. In Fig. 1C – F, we plot the values of these parameters for individual calls emitted by 8 animals, showing the extent of within- and between-individual variability over which generalization is required for *twitter* categorization. Similar generalization is required for categorizing the other call types as well (Supplementary Fig. 1).”

I. 71 “increasing resources” I guess you mean “increasing along the cortical processing hierarchy”, or “along the superior temporal lobe”? Please clarify. Same in I. 104.

- We have clarified that this means “increasing resources along the cortical processing hierarchy” at both locations.

I, 71-74 I don’t see why behavioral salience of calls and call selectivity in the temporal pole suggest that call classification is “a critical first step”. One could even argue that – as selectivity does not really arise before the temporal pole – it’s NOT the first step.

Also in line 105, based on the same premises, you say that “call processing is a computational GOAL of auditory cortical processing”, so there seems to be a disconnect here.

- The fMRI signal captures the activity of populations of neurons in the spatial scale of several millimeters. When we observe that BOLD responses to calls (all calls combined) are higher than responses to spectrally-matched non-calls, this means that a large fraction of cortical neurons in the area are responding strongly to calls. But in the Sadagopan (2015) study, we find a gradient of increasing selectivity that reaches its highest level at the temporal pole. We infer this to mean that selectivity develops over the cortical processing hierarchy.
- Secondly, call selectivity is not the end result of ‘call processing’. Extracting caller information, information about the behavioral state of the caller, and what the call might signify about the state of the environment may all require populations of neurons to selectively modulate their responses to calls. Thus, the first step might be to detect the presence of a call in the environment and determine its category, following which the above information can be extracted.
- We agree with the reviewer that we did not clearly make these arguments in our original submission. We have now better explained our rationale:
- **Page 5 / Line 74:** “The behavioral salience of calls for marmosets^{4 - 8}, and the increasing resources allocated to the processing of calls along the cortical processing hierarchy¹⁷, suggest that call processing is a computational goal of auditory cortex. Call processing involves detecting the presence of calls in the acoustic input, classifying them into behaviorally relevant categories, extracting information about caller identity, determining the behavioral state of the caller, and developing situational awareness of the environment. Although a number of studies have described call-selective responses at various stages of the auditory pathway, there has been little investigation into how the auditory system goes about solving these problems, both at the algorithmic and mechanistic levels. In this study, we started with the premise that the detection and classification of calls into discrete call types is a critical first step that enables the above computations. Our overall question ...”

I. 110 please say that for the purpose of the paper a “feature” is a randomly selected rectangular segment of a cochleogram limited in time and frequency (and not, for example, a parameter similar to those discussed around I. 480)

- We have now stated so in **Page 8 / Line 136:** “For the purposes of this study, a ‘feature’ is a randomly selected rectangular segment of the cochleogram, corresponding to the

spatiotemporal activity pattern of a subset of auditory nerve fibers within a specified time window.”

I would also like to see a justification why the features were rectangular and not, for example, elliptical (or weighed elliptical, e.g. 2D Gaussian).

- The features are rectangular because this is the shape with minimal assumptions. A ‘feature’ selective neuron could be conceptualized as monitoring a small set of auditory nerve fibers for a spatiotemporal pattern of neural activity. A weighted elliptical or 2D Gaussian shape would imply that the weighting of the activity of individual nerve fibers changes with time. Although we are not precluding this possibility, it introduces additional assumptions to the model which we would rather avoid. We now state this in the text of the revision.
- **Page 37 / Line 778:** “Each feature thus corresponded to the spatiotemporal pattern of activity of a subset of auditory nerve fibers within a specified time window (magenta box in Fig. 2B). We used rectangular feature shapes rather than other shapes to minimize assumptions – for example, an ellipse shaped feature would imply that the weighting of individual auditory nerve fibers changes over time.”

I. 112 I think that the way in which you describe ‘merit’ here may be confusing, the one in I. 610 is clearer.

- We have replaced the description of merit as recommended (new location **Page 8 / Line 140**).

Fig.3 Are MIFs in this figure the first five MIFs from Fig. 4A? Please clarify.

- Yes, this has been clarified in **Page 11 / Line 194**.

I. 142-143 “xxx vs. rest task” vs. “xxx vs. all other calls” in I. 129-130 – I presume these are the same, please harmonize. Also “task” does not feel right in the context.

- All instances changed to “xxx vs. all other calls”, and ‘task’ has been removed.

I. 147 I do not see how these MIFs capture harmonic relationships

- The harmonics in the cochleagrams in our figures are faint and at 100% magnification, are difficult to see. We apologize, and we have removed the reference to harmonics from the revised manuscript.

I. 173 “neurons” please state that these are simulated neurons. Also line 183, state simulated responses.

- Done

I. 200 Maybe: Control simulations?

- Done

I. 261 and elsewhere: I believe it’s guinea pig, not Guinea pig.

- We have removed the capitalization of “guinea” throughout the manuscript.

I. 276 one OR more?

- Yes, this typo has been corrected.

I. 278 as with comparison to real neuron properties, why this is done using only twitters?

- MIF-based reconstruction is not a core claim of this manuscript, and we intended the reconstruction to be a proof-of-principle demonstration. This is why it was restricted to one call type, and is relegated to a supplementary figure.

I. 285/6, 663 isn't "cross-correlation at lag 0" simply correlation?

- Yes, the reviewer is correct. But as we have been using the "cross correlation function" in the manuscript to signify that the feature can be found at any time point in a call, we emphasized "at lag 0" to ensure that it is understood the comparison of original and reconstructed calls is just a correlation.

I. 303 please do not abbreviate to "max. norm."

- Expanded to "maximum NCC", with NCC defined earlier as the abbreviation for normalized cross-correlation.

I. 309 "recordings"?

- We have replaced "recordings" with "neural data".

I. 388 what is meant by "four binary classification tasks", did you run classification between two callers in four pairs? If yes, why not 1 vs. all like with call type?

- Please see answer to next question below.

I. 391 Please explain why a small number of MIFs should indicate that calls differed in frequency

- Consider the distributions of dominant frequencies of twitters shown in new Fig. 1E. Animal 1 rarely produces a twitter less than 6.8 KHz, whereas all of Animal 4's twitters are less than 6.8 KHz. Thus, power at any frequency less than 6.8 KHz signifies that the twitter was emitted by Animal 4. In this case, all of the random initial features that are entirely <6.8 KHz will 'detect' all of Animal 4's twitters and not detect any of Animal 1's twitters. During the greedy search procedure, all of these features will be considered redundant because they share identical 'hits' and 'false alarms', and we will be left with one feature for classification. This is why a small number of MIFs indicate that calls differ in frequency.
- We ran pairwise classification tasks for this reason - when some of the callers just differed in dominant frequency, the results were rather obvious. We decided to present an example pair that was challenging to our model, where frequencies were highly overlapping (Animal 2 vs. 7).
- Following the reviewer's question, we have now run a 1 vs. all caller identification task, and have **replaced Fig. 10A** with these data. As you can see, we obtain much better performance, for the reason that proportionally, the number of 'hard' trials in this task is reduced compared to the earlier one, and the number of 'easy' trials is increased. We have moved the older example to new **Supplementary Figure 7**.

- **Page 26 / Line 527:** “We applied the greedy-search algorithm to determine the MIFs for caller identification in a caller A vs. all other callers task (Fig. 10A). We found that similar to call categorization, caller identification could also be achieved using a small number of MIFs ($n = 4$). If caller identification was performed in a binary fashion (four classifications between two animals each), in half of these tasks, classification could be accomplished using less than 3 MIFs, indicating that the calls of these marmosets probably differed along the frequency axis. This is because if there are clear differences in dominant frequency (for example, Animal 1 vs. 4 in Fig. 1E), all features that lie in one animal’s frequency range will detect all of that animal’s calls and none of the other animal’s calls. During the greedy search procedure, these features will be considered redundant and reduced to a single feature. In the other half, more MIFs were required for caller identification, and in general, MIFs were larger than those for call-type classification. This is likely because the differences between twitters produced by these animals are smaller compared to the differences between call types and can only be resolved in a higher dimensional space. Thus, integration over more frequencies and a larger time window may be necessary to resolve caller differences. In Supplementary Fig. 7, we plot the ROC for caller identification between a pair of marmosets with overlapping dominant frequencies. The MIF-based approach ($n = 20$ MIFs) achieved $>80\%$ hit rates with $<10\%$ false alarm rate for caller identification.”

I. 482 Your study also uses different sets of parameters (“features”) for different calls

- The difference with our study is that the ‘parameters’ in our case are frequency, bandwidth, and length of an arbitrary feature. That is, it is possible to search for a feature belonging to any call type in the other call types. We intended to contrast this with a parameter such as ‘inter-phrase interval’, which exists for twitters but is undefined for other calls. We have explained as follows, and would be happy to rephrase this if the reviewer has a specific suggestion.
- **Page 32 / Line 652:** “In our study, different MIFs are used for classification of different call types, but MIFs are parametrized along the same axes – bandwidth and integration window, allowing for a uniform basis for comparisons.”

I. 504 “non-mouse species” is rather awkward

- We have rephrased the two instances to “species other than mice” and “species with spectrotemporally complex calls”.

Reviewer #3 (Remarks to the Author):

My opinion is that the value of this manuscript is rather conceptual to find/use the informative acoustic features. I am somehow skeptical about whether it really is the neuronal mechanisms used in every species. On the other hand, their analyses may work well and bring similar conclusions for other well studied vocalizations like bat emission for echo-location or human phonemes.

- We now provide additional neural data (**new Fig. 9**, also please see responses to Reviewer 2 for data that we intend to publish in the near future), and hope that this alleviates some of the reviewer's skepticism. To expand upon the applicability of the model for other species, we have updated it to demonstrate the successful classification of multiple guinea pig call types and macaque vocalizations (**Fig. 10**). Here we were limited by the availability of large datasets. Since the original submission, our laboratory has recorded additional guinea pig vocalizations that we have used. We requested Dr. Yale Cohen for macaque vocalizations which he very generously provided (he is now acknowledged in the manuscript). We could not find large databases of the vocalizations of other animals, including bats. Our lab is in the early stages of a collaborative project for applying the model to classify some human speech sounds.
- **Page 27 / Line 549:** "We used the MIF-based approach to classify guinea pig call types ('whine', 'wheek', and 'rumble') from all other guinea pig call types. Similar to marmosets, guinea pig classification could be accomplished using a handful of features (12, 9, and 3 MIFs for whine, wheek, and rumble), and MIF-based classification achieved high performance levels (Fig. 10B). Similarly, we implemented the MIF-based algorithm to classify macaque calls (using 5, 4, and 9 MIFs for coos, grunts and harmonic arches) from a limited macaque call data set³³ and achieved high classification performance (Fig. 10C)."

Even though the approach to address vocal call coding is novel, and appealing well to the topic of vocal coding, there are several concerns that should be clarified.

(1) Are those spectrograms starting at the onset of vocal sound? It is not specifically described. But I believe that it is very important to mention for this study.

- Please see answer below

Assuming those started at the onset of sound, were durations of sounds similar to one another, between and within call types or callers? If durations differ systematically, e.g. 800 ms vs. 1 s, then a feature of 200 ms starting after 800 ms can be very informative.

- Please see answer below.

(2) For any classification tasks in the manuscript, I am not sure of the significance of any features (MIF) that start late. I guess that animals can find the call type (e.g. twitter or not) or caller identity (e.g. baby or adult) by the time such late features start. What are the onset timing of all primary features that are most informative?

- We apologize for our lack of clarity that has led to all three of these questions. The critical point here is that the features are restricted to occur at the same frequencies, but can occur anywhere in time within the call. The normalized cross correlation function essentially takes the feature and 'slides' it over the entire call duration, looking for matches at any time point in the call. In the original manuscript, we had tried to explain this by stating in the Methods: "We defined the 'response' of a feature to a call as the maximum value of the normalized cross-correlation function between the feature's cochleagram and the call's cochleagram,

restricted to the auditory nerve fibers that are represented in the feature.” In retrospect, we agree that this does not make clear that the feature can be found anywhere in time within a call.

- To clarify, we have now added to **Page 38 / Line 790**: “We effectively implemented a one-dimensional version of NCC by only considering the auditory nerve fibers that overlapped between the call and the feature. Note that this means features can only be detected in the frequency range that they span, but can be detected anywhere in time within a call.”
- Therefore, to answer the reviewer questions above, (1) it does not matter where the spectrograms start, feature detection happens even in the case of continuous inputs (such as the simulation in Fig. 5), (2) the durations do differ systematically between the calls, but it does not matter if a feature was originally extracted 800-1000ms after onset of a call because it can match to a segment that is at any time point in other calls, and (3) there is no particular significance to features that start late, and we did not find any relationship between onset timing and informativeness of the feature.

(3) Any bias in vocal call set, like the number of calls per animal subject, separately for each call type? It is desirable if the numbers of calls are nearly even across animals. Certainly, more subjects are better to obtain more general MIF. Thus, it is understandable if there is a bias of the number of calls between animals because the number of animals is limited. But it is better to clarify the compositions of data samples.

- The reviewer brings up some important points here that made us realize that our description of the training and test data sets and our definition of ‘production variability’ was unclear in the original manuscript. To answer these questions:
- We chose an almost equal number of calls per animal (about 60 calls/animal, 8 animals, 500 total) for each call type, both in the training and test sets. Indeed, access to a large database (from Xiaoqin Wang) was the main reason we used marmoset calls for developing and testing our model.
- In **new Fig. 1C-F** and **new Supp. Fig. 1**, we now characterize all marmoset calls used in the manuscript. In these figures, each individual dot corresponds to one call from one animal, and no obvious bias is observable in terms of sampling.
- The number of calls is now specified in **Page 37 / Line 768**: “...(about 60 calls per animal)...”
- We explain the composition of the data set in **Page 7 / Line 129**: “Marmoset twitters can be characterized along several acoustic parameters such as bandwidth, duration, dominant frequency, and inter-phrase interval⁸. In Fig. 1C – F, we plot the values of these parameters for individual calls emitted by 8 animals, showing the extent of within- and between-individual variability over which generalization is required for *twitter* categorization. Similar generalization is required for categorizing the other call types as well (Supplementary Fig. 1)”

The most informative MIF of twitter looks like one particular call, while other twitters apparently look different, making me worry that the number of calls of twitter is dominated by one animal that was making twitter call look like the MIF.

We think the reviewer is referring to Fig. 3. In Figure 3, we have merely outlined the MIFs on the ‘parent’ calls from which they were derived (Fig. 3 legend). For this ‘parent’ call alone, the normalized cross-correlation value between the feature and call is exactly 1. But because we optimize a threshold value for feature detection, this feature need not match the other calls

exactly – as long as the cross-correlation value crosses this optimal threshold, a ‘hit’ will be detected. In addition, as mentioned above, the MIF need not occur at this exact time – the cross correlation procedure will detect a peak no matter when in time the feature occurs.

- The larger worry of the reviewer is that we have over-trained our model on a particular animal, and are testing on a biased set. To address this point, as well as to demonstrate the extent of production variability over which the model must generalize, we have now added new analyses in **new Fig. 1C-F** and **new Supplementary Fig. 1** showing the distribution of call parameters for individual animals as well as how they vary in the training and test sets. The parametrization and analysis is based on methods developed by Agamaite et al. (2015). It is evident that there is high variability both within- and between-animals, that all animals are about equally represented, and there are no biases between the training and test set of calls used.
- **Page 7 / Line 129:** “Marmoset twitters can be characterized along several acoustic parameters such as bandwidth, duration, dominant frequency, and inter-phrase interval⁸. In Fig. 1C – F, we plot the values of these parameters for individual calls emitted by 8 animals, showing the extent of within- and between-individual variability over which generalization is required for *twitter* categorization. Similar generalization is required for categorizing the other call types as well (Supplementary Fig. 1).”
- **Page 13 / Line 235:** “This ‘test’ call set did not significantly differ from the training set along any of the characterized parameters (red histograms in Fig. 1).”

Authors could test how exclusion of one animal’s calls from dataset alters the MIF for call type classification task, how general the MIF is across animals.

- Please see answer below.

(4) Similar to my comment just above, while authors examined the performance of MIF by cross validation using different dataset between learning and testing, it would be nice if authors did so also by using features of a group of callers to classify call types of different callers.

We tested the model separately for sets of 4 randomly chosen animals and did not find any qualitative differences in the structure of features, or number of features. Merit of the top MIF was in the same range. This is now stated in the text. We have also added the suggested control where we train the model for twitter categorization on calls from 4 randomly chosen animals and test the model on 4 other animals. The ROC curve is plotted in **updated Fig. 6D** (triangles, labeled “Ind. Train/test”). Similar performance as the default model was obtained.

Page 16 / Line 305: “Twitter MIFs were not qualitatively different when derived from calls emitted by a smaller set of animals (4 animals). Training on a set of 4 animals and testing on 4 other animals yielded high performance (Fig. 6D), confirming the robustness of using MIFs for categorization of new calls.”

(5) When MIF differs in duration between datasets of call types, as seen in Fig 4, that may somehow relate to the temporal envelop modulation of sound. Isn’t it worth commented in text?

- This is related primarily to the durations of the calls (~1.25 s for twitters and phees, ~0.5 s for trills), as the MIF durations are similar when expressed relative to call length. But they indeed might be related to temporal modulations. We have commented in text as follows:
- **Page 12 / Line 217:** “In time, MIFs showed integration windows of the order of hundreds of milliseconds (Fig. 4B). The mean MIF lengths were 215 ms, 68 ms, and 406 ms for twitters,

trills, and phees respectively. Compared to the average lengths of the calls (twitters: 1.25 s, trills: 0.5 s, phees: 1.27 s), these correspond to 17%, 14%, and 32% of mean call length respectively. Interestingly, these lengths may correspond to time scales of temporal modulations in calls – for twitters, the sum of mean phrase length and mean inter-phrase interval is ~190 ms; for trills, the mean amplitude modulation period is ~30 ms. Thus, features of intermediate lengths were especially informative for call classification.”

(6) Figure 8. The title of the figure is “general applicability ... for other auditory tasks.” But I am not sure if it is fair enough to say “general.” “other auditory” are limited to caller identity or another species. Just “applicability ... for other tasks” sounds more consistent with the manuscript sense to me.

- We have changed Fig. 8 (now Fig. 10) legend to “The applicability... for other tasks” as requested.

Do authors already have actual neuronal responses that can classify calls?

- As we explained in our response to reviewer 2, this question of whether there are actual neuronal responses that can classify calls is precisely the current active area of research in our lab. These data have been presented at conferences (SfN 2017, Gordon Conference 2018, to be presented at ARO 2019), and we estimate that these experiments are about 70% complete, but we would require significant additional time to complete the analysis. Therefore, while we have decided to include a few more single neuron examples of feature-selectivity supporting the model in this revision, as we will explain below, for us to properly describe the experiments and necessary controls, we would need at least 4 additional figures and 2-3 more pages of text, which would complicate an already rich manuscript (10 main figures and 7 supplementary figures in this revision).
- Recall that at the MIF level, false alarms and missed detections are possible, which is why individual MIFs do not have 1 bit of information. True call categorization occurs by taking a weighted sum of MIF responses. In recordings from both marmoset and guinea pig A1, we do find neurons that appear to be selective for features across 1 or 2 call categories. Three examples from each species are shown in new Fig. 9.
- **Page 24 / Line 483:** “Consistent with the prediction of feature selectivity, we have found neurons in A1 of both marmosets and guinea pigs that respond selectively to conspecific call features. In Fig. 9, we present the spike rasters of example single neurons in both marmoset and guinea pig A1 responding to marmoset (Fig. 9A) and guinea pig calls (Fig. 9B) respectively. We presented multiple exemplars of each call type as stimuli. These example neurons responded at specific time points to a few call stimuli, typically across 1 – 3 categories. Such responses are consistent with our feature-based model because single features alone do not completely categorize calls, i.e., MIFs do not have 1 bit of information for categorization. Rather, combinations of features weighted by their log-likelihood ratios are necessary to ultimately achieve complete call category information. These data provide promising support for our model, but further experiments are necessary to: 1) determine how informative these neural features are about call category and how they compare with model features, 2) to confirm where such responses arise in the auditory pathway, and 3) to account for possible low-level confounds. Experiments are presently ongoing to address these issues.”
- Here, we also briefly summarize our ongoing experiments, not to be included in the manuscript, which show additional data for single neurons ability to classify calls.

- [Redacted text block]

| [Redacted text block]

| [Redacted text block]

| [Redacted text block]

- [Redacted text block]

[Redacted text block]

(7) Caller identity difference may relate more to F0. I may guess narrower bandwidths of MIF. Authors could do plots like Figure 4B for the MIF derived for the task in Figure 8A, if it reveals task-dependent differences of MIF.

- This certainly happens in our dataset, as is evident from new Fig. 1E. The caller identification example we had previously presented was between two animals that did not significantly differ in the F0 of the vocalization. We chose such a pair because when animals do differ in F0, we a single MIF is sufficient for classification – independent of any spectrotemporal structure, all that

is necessary is power in different frequencies. The MIF bandwidth in fact need not be narrow – for example, if the lowest frequency emitted by caller A is 4 KHz and caller B is 5 KHz, any power in the range 0 – 5 KHz would mean the caller is A. A feature can thus span 0 – 5KHz and have merit=1 for categorization of caller A. Thus, a plot like shown in Fig. 4B is not particularly informative, as it is reduced to a single point for some pairs.

- In this revision, as suggested by Reviewer 2, we present an example of one animal vs. all others for caller identification (updated **Fig. 10A**). In this set, there are some animals that overlap in F0 and others that do not. In this case, an F0 difference cannot be relied upon for identifying one caller from the other 7 marmosets, and MIFs that extract some higher level features are indeed necessary. The previous example is now presented in **Suppl. Fig. 7**.
- **Page 26 / Line 527:** “We applied the greedy-search algorithm to determine the MIFs for caller identification in a caller A vs. all other callers task (Fig. 10A). We found that similar to call categorization, caller identification could also be achieved using a small number of MIFs ($n = 4$). If caller identification was performed in a binary fashion (four classifications between two animals each), in half of these tasks, classification could be accomplished using less than 3 MIFs, indicating that the calls of these marmosets probably differed along the frequency axis. This is because if there are clear differences in dominant frequency (for example, Animal 1 vs. 4 in Fig. 1E), all features that lie in one animal’s frequency range will detect all of that animal’s calls and none of the other animal’s calls. During the greedy search procedure, these features will be considered redundant and reduced to a single feature. In the other half, more MIFs were required for caller identification, and in general, MIFs were larger than those for call-type classification. This is likely because the differences between twitters produced by these animals are smaller compared to the differences between call types and can only be resolved in a higher dimensional space. Thus, integration over more frequencies and a larger time window may be necessary to resolve caller differences. In Supplementary Fig. 7, we plot the ROC for caller identification between a pair of marmosets with overlapping dominant frequencies. The MIF-based approach ($n = 20$ MIFs) achieved >80% hit rates with <10% false alarm rate for caller identification.”

(8) If I adopt authors’ arguments about MIF representation in A1, I speculate that there are A1 neurons equipped to code MIF. Since cochleaographic representations of MIF are somehow equivalent to STRF, it would be nice to see some STRF measured for A1 neurons have similarity to MIF or partial overlaps with MIF. Similarly, I guess there are neurons tuned to lower frequencies in the region where MIF has vacant power as in the figure 3C, 2nd column. It would be nice to know whether such actual neurons, not MIF, remained silent during call sounds.

- We agree with the reviewer, and this is indeed what we are currently working on, as we have explained earlier to the reviewer’s question about whether we have found call selective responses.

(9) When simulated firing by MIF (Fig. 5), what timing were the MIF spike time relative to the period of MIF? Timing could be onset, center or offset of MIF. The durations differ between MIF. So, do long MIF need to wait longer to fire a spike? Are there A1 neurons that respond to calls in such way, too? Some MIF seem well exclusive to particular call type. Do actual neurons behave that way? This consideration may be more difficult when it comes to caller identity. Do some actual neurons respond exclusively to particular caller?

In the case of the simulation in Fig. 5, we assigned a spike to a simulated MIF neuron at the instant that the normalized cross correlation (NCC) value crossed its threshold. Because of how NCC is computed, this corresponds to the center of the MIF.

However, the reviewer is absolutely correct that neurons would have to 'wait' until a feature sufficiently matching its template would have to occur before firing. This is also currently being tested using electrophysiological recordings, as we have explained earlier. For example, when we extract 'neural' features from response events, we are investigating what timing of the stimulus relative to the response would yield the highest merit values. We predict that the feature entirely occurring before the response would yield the highest merit.

In macaque temporal pole there are suggestions of caller-selective neurons (the work of Perrodin and Petkov), but we have not encountered such neurons in marmoset at the level of primary auditory cortex. We do not yet have recordings from marmoset temporal pole.

(10) Random features extraction. This question is somehow obsolete, not affecting conclusions. But, it was not clear how. It seems every feature was a rectangle area of cochleograms, it would be defined by 4 variables (2 for position and 2 for size). I can imagine that such 4 variables can be swept systematically, from single pixels to the whole, to find most informative rectangle. Does random generation of rectangles mean random selections of those variables independent of one another?

- Yes, for each random feature, we chose a random center frequency, bandwidth, start time, and length independently. This is now defined better in the text.
- **Page 37 / Line 776:** "We then extracted 6000 random features from these 500 within-class cochleograms. To do so, we randomly chose a center frequency, bandwidth, onset time and length and extracted a snippet of activity from the cochleogram. Each feature thus corresponded to the spatiotemporal pattern of activity of a subset of auditory nerve fibers within a specified time window (magenta box in Fig. 2B)."

(11) Figure 7C. It is interesting to see peaks of firing rate response at 60 and 100 ms and a trough at 80 ms for neurons (top), and MIF (bottom) share such peaks and appear scarce at the trough. Any reason why?

- The reviewer's observation is correct – while the data shows a trough at 60ms, the simulated MIF responses do not. Two of the MIF responses (the ones with a peak rate of ~4 spk/s) do seem to hint at two peaks, but a clear trough is not evident. We do not have an explanation for this at the moment.

(12) Figure 4A. I think mutual information (merit) and log-likelihood ratio (weight) usually are monotonically related. Correct?

- As we mentioned in our responses to Reviewer 2, we realize that we have not clearly distinguished between 'merit' and 'weight'. Merit and weight need not be monotonically related. For example, it is possible that the lack of power in a certain frequency band is highly informative of a particular call type. A feature in this frequency band would have high merit for classification because it will reliably not be detected in the target category. But the feature would have a negative log-likelihood ratio as it is found more often in 'other' calls and not in the target category. We now state this in the Methods section.
- **Page 39 / Line 817:** "Note that merit and weight are distinct quantities that need not be monotonically related. For example, if the *lack* of energy in a frequency band is indicative of a target category, features that contain energy in this frequency band will be detected often in the other categories, but not in the target category. The feature will thus have high merit for classification, as it is informative by its absence, but have a negative weight."

- We also use merit and weight for different purposes – merit is used for the selection of MIFs, whereas weight is used for simulating MIF responses. This is because once a feature is chosen, we are only interested in the amount of evidence a spike provides as to the presence of a category, and weight is a more direct metric of this.

Am I correct about my description below?

“Added information are small for MIF after the first one. It is because the first column shows additional information after MIF in rows above added information. However, mutual information of MIF, that could be between 0 and 1, are high compared to added information, because many MIF have redundant information. Larger LL-ratio means the mutual information being informative is significant.” I think text could describe such details of how to interpret those parameters.

- The reviewer is correct for the most part, except the point about the LL-ratio (please see response to earlier question above). We agree with the reviewer that this could be better explained in the main text, and have done so in the revision.
- **Page 11 / Line 203:** “For twitters, detecting a single feature (the top MIF) was sufficient to gain 0.95 bits of information. Subsequent features probably detected only a few additional twitters without introducing new false alarms. For the other call types, however, the top MIF only provided 0.78 or 0.6 bits of information. Although successive MIFs individually had high merit (second column), they added little information to the top MIF (first column), likely because of redundancy – each MIF could only add a small number of additional ‘hits’ without introducing new false alarms. However, detecting these features was crucial for solving the task, as they ultimately elevated the total information to > 0.9 bits. The MIFs have positive weights, suggesting that they are informative by virtue of their presence (rather than absence) in the target category.”

When MIF position/size differed, how redundant are they about mutual information?

- We now show the effect of including or excluding MIFs of different sizes on classification performance in **new Fig. 7**. We did not find a systematic dependence of MIF position in time on merit. In frequency, MIFs that were positioned over frequency bands in which the call had power had higher merits.

(13) Several words deserve more explicit descriptions.

What is “normalized cross correlation”? Sometimes it is not clear cross correlation between what. It is entirely not clear what was normalized by what. Only description of it in the text is “commonly used metric to quantify template match.”

- We apologize for the unclear definition. Regular cross-correlation yields values that are dependent on the intensities of the signal in the two patches (of images or cochleagrams) being correlated. When computing cross-correlation of patches that have changes in intensities, this means that the highest cross-correlation value may not necessarily correspond to the best match. A second problem is that the regular cross-correlation value is not bounded. To avoid these problems, the intensities of the patches being correlated are power-normalized by subtracting their means and dividing by their standard deviations before convolution. This is the normalized cross-correlation (NCC), and this quantity is bounded between -1 (completely anti-correlated) and +1 (perfect match). This is now explained in the methods:
- **Page 38 / Line 787:** “*Threshold optimization:* We defined the ‘response’ of a feature to a call as the maximum value of the normalized cross correlation (NCC) function between the feature’s cochleagram and the call’s cochleagram, restricted to the auditory nerve fibers that are represented in the feature. We effectively implemented a one-dimensional version of NCC by

only considering the auditory nerve fibers that overlapped between the call and the feature. Note that this means features can only be detected in the frequency range that they span, but can be detected anywhere in time within a call. NCC is a commonly used metric to quantify template-match. To compute the NCC, the feature and the cochleagram patch at each lag were normalized by subtracting their respective mean values and dividing by their respective standard deviations before convolving them. This results in a value between -1, signifying that the feature and cochleagram patch at that lag are completely anti-correlated, and +1, signifying a perfect match between the feature and the cochleagram. Because this is a computation-intensive step ...”

What is “power law nonlinearity”? It could be anything with $y=Ax^n$ with any n other than $n=1$. Texts in Results direct readers to see Methods. But, in Methods, it is said only “that was scaled arbitrarily”. What was scaled arbitrarily is not clear either. The power? The proportionality?

- We apologize for the unclear definition. We fixed the exponent at 4 ($n=4$), and arbitrarily scaled A to match the data. This is now stated in the Methods:

Page 41 / Line 861: “These were converted to firing rates by applying a power law nonlinearity, of the form:

$$FR = k \cdot [V_m - \theta]^p$$

Where FR is the firing rate response in spk/s, θ is the MIF’s optimal threshold, p is the exponential nonlinearity set to a value of 4, and k is an arbitrary scaling factor.”

(14) Figure 7 legend. “B-top” should be “A-top,” and “C-E” should be “B-D.”

- Thank you for pointing out these errors. Figure 7 legend has been corrected.

(15) “one of more of MIF”

- This typo has been corrected.

Reviewers' Comments:

Reviewer #1:

Remarks to the Author:

The authors satisfactorily responded to my concerns. I recommend the paper for publication.

Reviewer #2:

Remarks to the Author:

I am completely satisfied with the responses and changes by the authors and have no further comments or requests.

Reviewer #3:

Remarks to the Author:

Authors clarified all my concerns. The manuscript is revised well. I have no further comments.

Response to Reviewers

We thank the reviewers again for their time and significant contribution to the manuscript.

Reviewer #1 (Remarks to the Author):

The authors satisfactorily responded to my concerns. I recommend the paper for publication.

Reviewer #2 (Remarks to the Author):

I am completely satisfied with the responses and changes by the authors and have no further comments or requests.

Reviewer #3 (Remarks to the Author):

Authors clarified all my concerns. The manuscript is revised well. I have no further comments.